



# Sensitivity of iceberg drift and deterioration simulations to input data from different ocean, sea ice and atmosphere models in the Barents Sea (Part II).

Lia Herrmannsdörfer[1], Raed Khalil Lubbad[1], and Knut Vilhelm Høyland[1]

[1]Norwegian University of Science and Technology, Trondheim, Norway

**Correspondence:** Lia Herrmannsdörfer (lia.f.herrmannsdorfer@ntnu.no)

**Abstract.** Iceberg data in the Barents Sea is scarce. Numerical simulations of iceberg drift and deterioration as function of the environmental conditions, e.g. from models of atmosphere, ocean and sea ice, provide a useful mean to bridge this gap. The simulation results rely on the quality of the input data. We conduct a numerical experiment, in which we force an iceberg drift and deterioration model with combinations of two atmospheric reanalyses (ERA5, CARRA) and two ocean and sea ice models (Topaz, Barents-2.5) in the Barents Sea and the years of 2010-2014 and 2020-2021. Further, the impact on the simulation results is analysed. We found that simulation results of iceberg drift and deterioration are sensitive to the choice of the ocean and sea ice forcing data. The horizontal resolution bathymetry of the forcing data, especially in proximity to the coastlines, influence the availability and representability of the forcing information and, thus, the iceberg simulation results (e.g. occurrence and extent). Deviations in the ocean and sea ice variables in Barents-2.5 and Topaz caused considerable differences in the simulated large-scale and regional iceberg occurrence in the domain. The impact is especially large for sea ice variables. The impact of varied atmospheric forcing is secondary. In spite of varied environmental forcing, surprising similarities in the main iceberg pathways were observed.

## 1 Introduction

The Barents Sea is subject to icebergs calved from the tidewater glaciers of Svalbard, Franz-Josef-Land and Novaya Zemlya (Abramov and Tunik, 1996). The number of iceberg observations is limited due to their comparably small size, rare occurrence in the more inhabited southwestern Barents Sea and the general sparseness of observations in the Arctic. Statistics of iceberg occurrence therefore rely on numerical simulations.

Such simulations describe the interaction of ice features with the atmosphere, ocean and sea ice. Iceberg drift is steered by the sea water motion, waves, wind, sea ice drift, sea surface slope and coriolis forces (Savage, 2001). Icebergs deteriorate by wave erosion, calving, forced convection by sea water and wind, buoyant vertical convection and solar radiation (El-Tahan et al., 1987; Savage, 2001). Previous studies (Kubat et al. (2005, 2007); Eik (2009b, a); Keghouche et al. (2009, 2010)) investigated the relative importance of those parameters and validated their model implementations with observational and experimental data. Eik (2009b) highlighted the influence of the environmental forcing and concluded that a large part of the uncertainties



resolves from the environmental input, e.g. from ocean reanalyses.

Due to its diverse bathymetry and position between warm Atlantic waters and the cold Arctic Ocean, the Barents Sea exhibits highly complex interaction of ocean and atmosphere. Those interactions are described in atmospheric, ocean and sea ice models, with different resolution and physical description.

This paper is dedicated to study the impact of varied ocean, sea ice and atmosphere forcing on simulations of iceberg drift and deterioration in the Barents Sea. The results shall improve the accuracy of iceberg predictions and statistics in the Barents Sea and shall also support the choice of environmental forcing for iceberg simulations.

Therefore, a stat-of-the-art model for the simulation of iceberg drift and deterioration in the Barents Sea described by Monteban et al. (2020) is used herein to perform a numerical experiment. In the experiment, the iceberg model is forced by various combinations of ocean and atmosphere reanalyses, hindcasts and forecast systems, of different resolution and representativity of the domain, namely *Arctic Ocean Physics Reanalysis (Topaz)* (MDS, 2023), the Barents-2.5 forecast system (MET-Norway, a), the Barents-2.5 hindcast (MET-Norway, b), the *global atmospheric reanalysis ERA5* (Hersbach et al.) and the *Arctic re-*
*gional reanalysis CARRA* (Schyberg et al.).

A total of 72884 icebergs ($4 \cdot 2603 \cdot 7$) is simulated and the results are analysed statistically (Sect. 3.1-3.5). Further we examined one exceptional iceberg trajectory (Sect. 3.6). The differently-forced simulations are compared regarding various characteristics of the simulated iceberg trajectories, i.e. the availability of forcing data in the iceberg model's data assimilation
(Sect. 3.1), iceberg deterioration (Sect. 3.2), iceberg drift (3.3) and resulting distribution in the domain (Sect. 3.4, 3.5).
   In the discussion (Sect. 4), the differences of the simulations with varied forcing are traced back to the differences of the forcing variables and their origin in the setup of the ocean, sea ice and atmosphere models, described in the preceding study, Herrmannsdörfer et al. (2024a).

## 2   Description of the Experiment

A numerical experiment is conducted in which the model of Monteban et al. (2020) for iceberg drift and deterioration is forced by different combinations of ocean, sea ice and atmosphere data sets to determine the impact of varied forcing. This Section provides an overview on the experiment, input data and the iceberg model. A detailed description of the iceberg seeding, the iceberg model setup, the drift and deterioration equations, the equation parameters and the computational routines are given in the Appendix (Sect. A).





**Table 1.** Combinations of environmental forcing from ocean, sea ice and atmospheric reanalyses, hindcasts and forecasts in the numerical experiment.

| Objective | Ocean & sea ice Forcing | Atmospheric Forcing |
|---|---|---|
| Reference, global | Topaz | ERA5 |
| Regional wind | Topaz | CARRA |
| Regional ocean & sea ice | Barents-2.5 | ERA5 |
| High resolution, fully regional | Barents-2.5 | CARRA |

## 2.1 Experiment setup

The iceberg model is forced by the four combinations of the ocean, sea ice and atmosphere data shown in Table 1. The forcing combinations represent a reference case with global forcing (Topaz and ERA5) and a high-resolution, regional simulation (with Barents-2.5 and CARRA). The other combinations serve to estimate the individual influence of ocean, sea ice and atmosphere forcing on the simulations results. We did not conduct a full sensitivity analysis for every variable, as this would cause physical inconsistency and does not reflect a probable use case. The simulations are performed for the years 2010-2014 and 2020-2021 (7 years), due to limitations in the data availability at the time the simulations were performed. Following, a total number of $2603 \cdot 7$ icebergs are simulated in four different forcing combinations.

## 2.2 Data

This Section provides an overview on used variables, data sources and applied pre-processing. A detailed description is given in Herrmannsdörfer et al. (2024a).

We use 10m wind ($v_\mathrm{a}$) from Global Atmospheric Reanalysis ERA5 (Hersbach et al.) and Copernicus Arctic Regional Re-Analysis CARRA (Schyberg et al.). The sea surface velocity ($v_\mathrm{w}$) and surface temperature ($SST$), as well as, the sea ice concentration ($CI$), thickness ($h_\mathrm{si}$) and drift velocity ($v_\mathrm{si}$) are obtained from the Arctic Ocean Physics reanalysis based on Topaz4b (MDS, 2023) (Topaz), Barents-2.5 forecast (2020-2022) (MET-Norway, a) and Barents-2.5 Hindcast (2010-2014) (MET-Norway, b). Further, geostrophic currents are gathered from Slagstad et al. (1990) and bathymetry is gathered from (Jakobsson et al., 2012).

To reduce memory requirements, a spatial subset of the Barents Sea is applied. Note, that the subsets differ for the forcing data sets, due to their varying grid type, resolution and orientation. The ERA5 and CARRA data is masked for grid cells with at least $50/75\%$ ocean content, based on their native land-sea-masks.



For an efficient simulation process, velocity variables are translated to components in the longitudinal (u) and latitudinal (v) direction. In addition, temporal data gaps on the scale of hours to few days are replaced by the previous or following time step for a more consistent forcing in the iceberg simulations.

### 2.3 Iceberg seeding

Icebergs are seeded at a random position close to Franz-Josef-Land, Austfonna, Edgeøya and Novaya Zemlya and a random day from 1 July to 30 November of the simulations years 2010-2014 and 2020-2021. The iceberg length is drawn randomly from a generalised extreme value (GEV) distribution, derived from satellite observations at each of the sources by Monteban et al. (2020). The iceberg width and height are derived by empirical relations. The minimum initial iceberg length is defined by $34\,\mathrm{m}$ corresponding to the maximum resolution of the satellite observations and the definition of a bergy bit ($10\,\mathrm{m}$ height). This initial conditions vary for all 2603 icebergs released within one simulation year and the 7 simulation years, but is reproduced in the simulations with varied forcing. More details on the seeding approach are given in the Appendix (Sect. A).

### 2.4 Model for iceberg drift and deterioration

The numerical model for the simulation of iceberg drift and deterioration is adapted from Monteban et al. (2020) to suite the requirements of this study. The model is Lagrangian and deterministic. The iceberg drift is simulated based on wind, sea water velocity, sea ice drift and the resulting coriolis force. The pressure gradient forces are approximated with the geostrophic currents from Slagstad et al. (1990). Wave drag forces are included in the wind drag coefficient implicitly. The added mass coefficient is set to zero. The iceberg melt is a function of basal turbulent melt, vertical thermal buoyant convective melt and wave erosion based on wind, water velocity and the sea surface temperature. The wave erosion term does not consider swell waves as the sea state is defined only by wind and water velocity. Melt by solar radiation is neglected and calving is not explicitly described. The drift and deterioration equations and model parameters are given in the Appendix (Sect. A).

The model solves the drift and deterioration equations for $2-\mathrm{hourly}$ time steps and updates iceberg position, size and velocity. The simulation is stopped when the iceberg is melted to the size of a growler ($H \leq 10\,\mathrm{m}$), leaves the simulation domain or time period.

### 2.5 Assimilation of environmental data

Environmental data is assimilated at $2-\mathrm{hourly}$ simulation time steps at the iceberg position. Dependent on the temporal resolution of the data, forcing fields are read directly at the $2-\mathrm{hourly}$ time steps (ERA5, Barents-2.5) or the last available time steps (Topaz, CARRA). Spatially, forcing variables are read from the nearest grid cell of the respective data set (*nearest forcing cell*) without interpolation, as shown in Fig. 1 (green circles). The distance is determined between the iceberg and the grid cell centre (Fig. 1, green lines). Note, that the environmental data sets have different grids, so that the forcing data for one time step





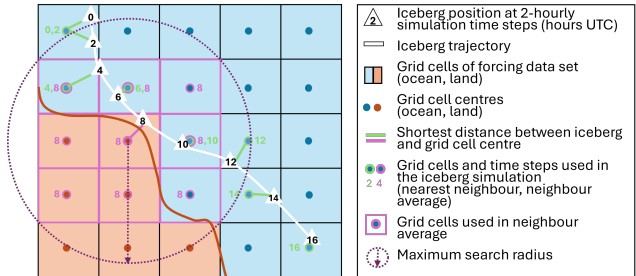

**Figure 1.** Spatial assimilation of gridded environmental data into the iceberg model. At every $2 - \text{hourly}$ simulation time step (numbers, triangles) of the iceberg trajectory (white line), the environmental data is assimilated from the nearest forcing grid cell (*nearest forcing cell*, green circles). The distance is estimated between iceberg and grid cell centre (green lines). If the nearest forcing cell does not contain data (e.g. over land), the average is calculated from the surrounding available forcing cells (*neighbour average*, pink lines, squares and circles). Note the depiction of the coastline (brown line), ocean (blue dots and background) and land grid cells (brown dots and background).

is not raised from the same area.

Because of the gridded nature of the forcing data and the respective resolution, ocean forcing is often not available close to the coastlines. As the iceberg model is Lagrangian, has a high-resolution bathymetry and a different representation of the coastline, icebergs may drift close to (or beyond) the coastline of the forcing data set. Figure 1 illustrates an iceberg trajectory, in which the iceberg determines a land cell as nearest forcing cell, in which ocean and sea ice variables are unavailable (Fig. 1, time step 10). In the current implementation of the iceberg model, missing forcing along the coastlines is approximated

by averaging surrounding grid cells ("neighbour average", Fig. 1, pink squares). If the surrounding grid cells of the nearest neighbour forcing cell do not contain any data either, the search radius (Fig. 1, purple circle) is increased step-wise, and up to a radius of 3 (for low-resolution input) or 16 grid cells (for high-resolution input), for every variable separately. If no forcing data is available within this radius, the respective forcing variable is set to zero. This corresponds to a maximum search radius of roughly $52\,\text{km}$ grid cells for Topaz and $57\text{km}$ for Barents-2.5 and CARRA. The maximum search radius in kilometres

$sr_{max}(km)$ is calculated by

$$sr_{\max}(\text{km}) = \sqrt{2 \cdot (d \cdot sr)^2} \tag{1}$$

with the horizontal grid resolution $d$ (e.g. $2.5\,\text{km}$ for Barents-2.5) and the search radius ($sr$) as number of grid cells (e.g. 1). In contrast, the nearest forcing cell exhibits a maximum search radius of $\sqrt{2 \cdot (d/2)^2} = \sqrt{2 * (12.4/2)^2} = 8.8\,\text{km}$ for Topaz, $1.8\,\text{km}$ for Barents-2.5 and CARRA. Note that ERA5 has a regular latitude-longitude grid (Hersbach et al., 2020) and the

search radius varies with the latitude accordingly.





## 3 Analysis

### 3.1 Availability of environmental forcing data in the data assimilation

The simulations of iceberg drift and deterioration are influenced by the availability of the environmental forcing data (Bigg et al., 1997; Kubat et al., 2005; Eik, 2009a) and data assimilation approach in the iceberg model (Herrmannsdörfer et al.,
2024b). In the following Section, the spatial availability of the forcing variables is analysed for the iceberg model setup (Sect. 2.4) and the different forcing datasets (Sect. 2.2).

The iceberg model assimilates environmental data at every $2-\text{hourly}$ simulation time step from the nearest grid cell of the forcing data grid (*nearest forcing cell*) or averages the surrounding grid cells (*neighbour average*) with an increasing search
radius $sr$ (Sect. 2.4, Eq. 1). We investigate the relative number of simulated iceberg trajectories and simulation time steps that used nearest neighbour or neighbour average of a certain radius. Note that we differentiate the usage (of e.g. neighbour average of radius 1) in the iceberg trajectories, and simulation time steps, aggregated from all simulated iceberg trajectories.

Figure 2 visualises the horizontal grid resolution of the forcing data sets, the search radius and how often it is used in Topaz-,
Barents2.5-, ERA5- and CARRA-forced iceberg simulations. The x-axis shows the maximum search radius $sr$, which represents the maximum distance of an iceberg to the closest grid cell centre with available forcing data. The y-axis shows how often forcing is aggregated at the maximum search radius on the x-axis. Note, that the percentage changes step wise along the x-axis corresponding to the respective grid resolution. The plot elements show the percentage of available data relative to the number of iceberg trajectories (bars), simulation data points (time steps aggregated from all trajectories) in which all variables
are available (lines) in the differently forced simulations (colours). Note that the numbers do not add up to $100\%$, as different variables, or time steps within a trajectory, are often acquired from different search radii.

*Availability of data in the nearest forcing cell*
In general, Fig. 2 shows, that in the majority of trajectories ($86\%$) and the majority of time steps ($98\%$), the forcing variable(s)
are available in the nearest forcing cell.

*Neighbour averaging*
The neighbour average is applied (due to missing data in the nearest forcing cell) in all differently forced simulations, however the frequency of occurrence and the search radius differ in the simulations with varied forcing. $36\%$ of Topaz- and $50\%$ of
Barents-2.5-forced trajectories, and $10\%$ of Topaz- and $14\%$ of Barents-2.5-forced time steps, require neighbour averaging for some variable and time step. Further investigation (not shown) indicated that the availability of Barents-2.5 variables differs due to the variable's native grid upon production.



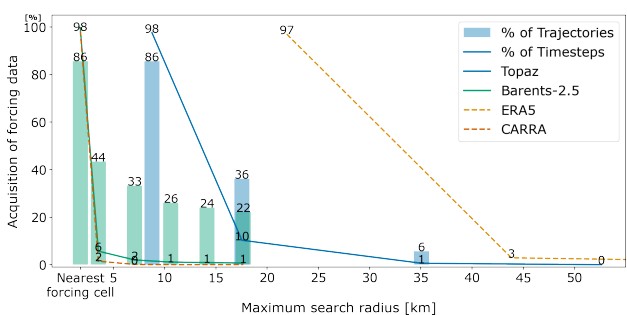

**Figure 2.** Acquisition (%, y-axis) of different environmental forcing data (colours) in the iceberg simulations and distance of acquisition (x-axis). The iceberg model assimilates environmental forcing from the forcing data's grid cell with the smallest distance to the current iceberg position (nearest neighbour approach, see Sect. 2.4 and Fig. 1). In the case of unavailable forcing in the nearest forcing cell, data is averaged from the surrounding grid cells of increasing search radius. The availability of forcing data (%, y-axis) is given as function of the maximum search radius (or maximum distance between iceberg and grid cell centre) (x-axis), that changes step wise dependent on the horizontal data resolution. The availability is given for the datasets of Topaz (blue), Barents (green), ERA5 (orange) and CARRA (red). The availability is given relative to the number of iceberg trajectories (bars) and simulation time steps (lines).

*Search radii*

For most of the neighbour averages, a search radius of 1 forcing cell is sufficient. Larger search radii are necessary with decreasing frequency. Around $6\%$ ($< 1\%$) of the Topaz and $34\%$ ($2\%$) of the Barents2.5-forced trajectories, but only $< 1\%$ of the Topaz and $2\%$ of the Barents2.5-forced time steps, use search radii larger than one cell ($12.3\,\mathrm{km}$ and $2.5\,\mathrm{km}$ respectively). The largest applied search radius was 3 forcing cells ($53\,\mathrm{km}$) for Topaz and 4 forcing cells ($14\,\mathrm{km}$) for Barents-2.5. Further studies (not shown) indicated that Barents-2.5 sea ice velocity is partly unavailable within a larger search radius.

*Availability of wind data*

The availability of wind forcing in the nearest forcing cell is higher than for the other forcing variables and is available within a search radius of 3 cells or less for both ERA5 (not given in km) and CARRA ($11\,\mathrm{km}$). Due to its regular latitude-longitude-grid, the search radius for ERA5 wind is limited by its coarser latitudinal resolution and is therefore given as function of the approximate resolution in the Arctic of $31\,\mathrm{km}$ in Fig. 2.

## 3.2 Iceberg deterioration

The iceberg deterioration accounts for wave erosion, buoyant vertical convection and basal melt in this setup (Sect. 2.4). The deterioration Eq. are described in detail in the Appendix.





**Table 2.** Contribution to iceberg deterioration from different forcing terms (wave erosion $M_{\mathrm{e}}$, basal melt $M_{\mathrm{fw}}$ and vertical buoyant convection $M_{\mathrm{v}}$) in Topaz- and Barents-2.5-forced simulations and their difference. The values are expressed as relative contribution to the total mass loss or deterioration ($\delta$, in %).

|  | $\delta(M_{\mathrm{e}})$ | $\delta(M_{\mathrm{fw}})$ | $\delta(M_{\mathrm{v}})$ |
|---|---|---|---|
| Topaz | 74 | 27 | 0.8 |
| Barents-2.5 | 55 | 43 | 0.6 |
| Topaz-Barents2.5 | +19 | -17 | +0.1 |

The total iceberg deterioration rate of simulation time step $j$ (with length $dt = 3600 \cdot 2\,\mathrm{s}$) is measured as mass loss or reduction in volume times the density $\rho_i$ of glacial ice (Eq. 2).

$$\delta = m_{j+1} - m_j = \rho_i \cdot (H_{j+1} \cdot W_{j+1} \cdot L_{j+1} - H_j \cdot W_j \cdot L_j) \tag{2}$$

$$= \rho_i \cdot ([H_j + \Delta H_j] \cdot [W_j + \Delta L_j] \cdot [L_j + \Delta L_j] - H_j \cdot W_j \cdot L_j) \tag{3}$$

where $\Delta H_j = M_{\mathrm{fb}} \cdot dt$ is the reduction in height during time step due to melt at the base ($M_{\mathrm{fw}}$). $\Delta L_j = \Delta W_j = (M_{\mathrm{e}} + M_{\mathrm{v}}) \cdot dt$ is the reduction at the sides due to wave erosion ($M_{\mathrm{e}}$) and vertical buoyant convection ($M_{\mathrm{v}}$).

We introduce the measure of the relative contribution of a melt term to the total deterioration ($\delta$, in %)

$$\delta = \frac{\sum_{j=1}^{J} \delta(j, term)}{\sum_{j=1}^{J} \delta(j)} \cdot 100\% \tag{4}$$

where $j$ is any simulation time step and $J$ is the number of simulation time steps of $2603 \cdot 7$ icebergs of similar environmental forcing. The contribution $\delta$ by wave erosion is largest ($55 - 74\%$), followed by basal melt ($27 - 43\%$), and vertical convection ($0.6 - 0.8\%$) (Table 2). However, the relative importance of the deterioration terms varies with the environmental forcing. Comparing the differently forced simulations, Topaz-forced icebergs have larger wave erosion ($+19\%$) and smaller basal melt ($-17\%$). The difference in vertical convection is small, as the relative contribution. Further studies (not shown) indicated a $6.2 \cdot 10^4\,\mathrm{kg}$ larger mass loss averaged over all time steps in Topaz-forced simulations. It also showed small melt rates for times steps in sea ice. An example of the relative contributions of the deterioration terms and their deviations for differently forced simulations is also given in Sect. 3.6.

## 3.3 Iceberg drift duration and distance

This Section examines how the drift duration and distance are influenced by the selection of ocean, sea ice and atmosphere forcing, for a range of seeding conditions. In detail, the distance along the trajectory (*Track*), the distance between seeding and the melt position (*Effective*) and the time, in which an iceberg persists until it is melted (*Duration*), are analysed.





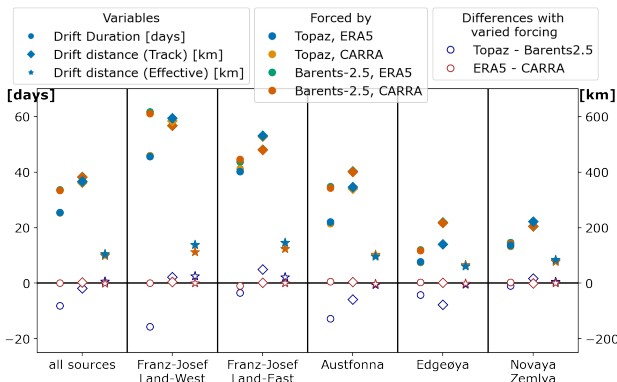

**Figure 3.** Average iceberg drift duration [days] and distance (Track, Effective) [km] and their difference under varied forcing (y-axis) for differently forced iceberg trajectories (colours), originating from the sources Franz-Josef-Land East and West, Austfonna, Edgeøya and Novaya Zemlya (x-axis).

Comparing simulations of different ocean and sea ice forcing, Topaz-forced icebergs drift in average $8\,\mathrm{days}$ shorter in time, $18\,\mathrm{km}$ less in distance along the track, but $6\,\mathrm{km}$ more in effective distance (Fig. 3). These differences are partly relevant as they make up $28\%$, $5\%$ and $6\%$ of the mean absolute values of all simulations. The difference between simulations with varied
atmospheric forcing are minor.

The iceberg drift duration and distance also vary from source and its seeding characteristics (Fig. 3). The dependency on the seeding location causes a variation of $8$ to $62\,\mathrm{days}$, $140$ to $594\,\mathrm{km}$ (Track) and $61$ to $146\,\mathrm{km}$ (Effective). Iceberg drift duration and distances (Track, Effective) are highest for icebergs originating from Franz-Josef-Land and Austfonna, and smaller for
Edgeøya and Novaya Zemlya in all simulations.

Analysing the dependency on both source and forcing, the differences between the sources dominate the difference due to the forcing (Fig. 3). Similarities and differences between the iceberg drift from various sources are to a large extent reproduced by the differently forced simulations. However, differences between Topaz- and Barents-2.5-forced drift are significant and vary in both sign and magnitude. The drift distance (Effective, Track) of Topaz-forced trajectories is longer for icebergs originating
from Franz-Josef-Land and Novaya Zemlya, while it is shorter for icebergs from Austfonna and Edgeøya. The drift duration is shorter for Topaz-forced simulations. The difference due to varied atmospheric forcing is minor.

### 3.4 Spatial iceberg density

Iceberg density is a measure to express the average number of icebergs in a domain over a time period, along with the number of simultaneous occurrences. In this study, the iceberg density is compared for the differently forced simulations to analyse the





spatial differences in iceberg drift and deterioration.

The iceberg density is derived by the number of icebergs *i* that are within a defined grid cell at the same simulation time step, the number of time steps *n* in which *i* icebergs are within the same grid cell and the total number of simulation time steps *N*. The probability of having *i* icebergs in one grid cell at same time step $p(i)$ is calculated for every occurring *i* by Eq. 5. The areal density $\rho_a$ (called *iceberg density* in the following) is given by Eq. 6 with the surface of the grid cell $A_{\mathrm{gridcell}}$. In this analysis, iceberg density is accumulated on an artificial grid of $25\,\mathrm{km}$ horizontal resolution.

$$p(i) = \frac{n(i)}{N} \tag{5}$$

$$\rho_a = \frac{\sum_{i=0}^{\infty} i \cdot p(i)}{A_{\mathrm{gridcell}}} \tag{6}$$

Figure 4 shows maps of the iceberg density in the Barents Sea for the differently forced simulations. The colour scale in Fig. 4 supports the data below the 95th percentile of the densities (grey, $2 \cdot 10^{-4}$) and the highest $5\%$ (orange colouring and white line, individual for every simulation). Iceberg densities are highest (largest $5\%$, white line and orange colour map in Fig. 4) around eastern Svalbard, Franz-Josef-Land and northwestern Novaya Zemlya (Sect. 2.4) and decreases with increasing distance to those locations, independent of the forcing.

Figure 5 shows the spatial density differences between the differently forced simulations. Density differences are as large as absolute densities, which is highlighted by similar density range in Fig. 4 and 5. Deviations by varied ocean and sea ice forcing are larger than deviations by varied atmospheric forcing, however some effects of ocean and atmospheric forcing add up while some cancel each other out.

Simulations of varied ocean- and sea ice-forcing show significant difference in large parts of the domain, and especially around Svalbard and Franz-Josef-Land (Fig. 5). Thereby, Topaz-forced simulations have higher density close to the coastline of Franz-Josef-Land and Svalbard, and lower density in larger proximity of the archipelagos. The density differences are decreasing towards the open ocean.

A selection of regional density difference from Fig. 5 is noted in the following. Iceberg densities are larger for Barents-2.5-forced simulations in the northernmost parts of the domain. Iceberg density is higher to the north and west (south) of the main iceberg source on Novaya Zemlya for Topaz (Barents-2.5)-forced simulations. For more, iceberg density is larger to the north-west (north-east) of Bjørnøya for Topaz (Barents-2.5)-forced simulations.



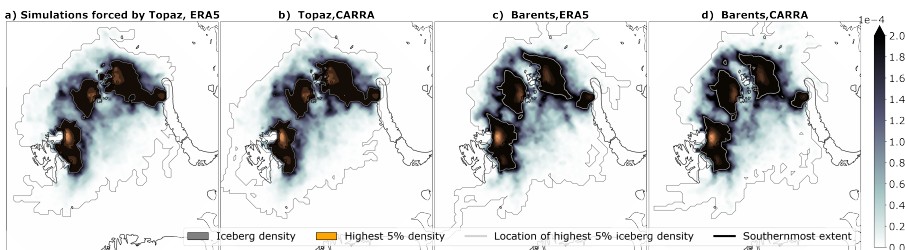

**Figure 4.** Iceberg density (colours) and southernmost extend (black line), aggregated for simulations forced by a) Topaz and ERA5, b) Topaz and CARRA, c) Barents-2.5 and ERA5 and d) Barents-2.5 and CARRA. The largest $5\%$ of the respective simulations' iceberg density is highlighted (white line, orange colour map).

## 3.5  Iceberg extent

The iceberg extent is a measure of how far icebergs drift, how much they spread and how much they are restricted to common pathways. The spatial iceberg extent is shown by black lines in the Fig. 4. All simulated icebergs in the accumulated time period 2010-2014 and 2020-2021 drifted within the indicated lines. Icebergs in Barents-2.5-forced simulations show larger spread in the domain, in all directions (Fig. 4). In contrast, Topaz-forced icebergs drift further north and west from Svalbard.

More detailed analysis (not shown) indicates that the southernmost iceberg trajectories reach to the south of Bjørnøya and to the south-eastern Barents Sea around $72-74°$N, independent on the forcing. Some of those icebergs drift within sea ice of high concentration and some drift in open waters. The southernmost trajectory reached $72°$N in the Central Basin for the Barents-2.5-forced simulations and is described in Sect. 3.6.

The iceberg extent can also be described by the relative number of grid cells that contain icebergs at a given time step, the iceberg extension. In the following, the iceberg extension is analysed as time series from 2010 to 2014 and 2020 to 2021 (Fig. 6). The iceberg extension varies in time and between the differently forced simulations (Fig. 6). The iceberg extension varies for simulations with different atmospheric and ocean-sea ice forcing, and at a similar scale.

In detail, the iceberg extension has a seasonal cycle and multi-year variability (Fig. 6). This variability is reproduced in all differently forced simulations, however with small deviations. The iceberg extension increases from July to December, when icebergs are seeded, and decreases again until July. The period of July to December is characterised by large deviation with varied ocean forcing and small variations with atmospheric forcing. Largest differences with varied ocean and sea ice forcing occur in August to September and November to December. The period from December to June shows similar deviations with varied ocean and atmospheric forcing. These deviations are differently pronounced in the individual years of the timeseries (Fig. 6).



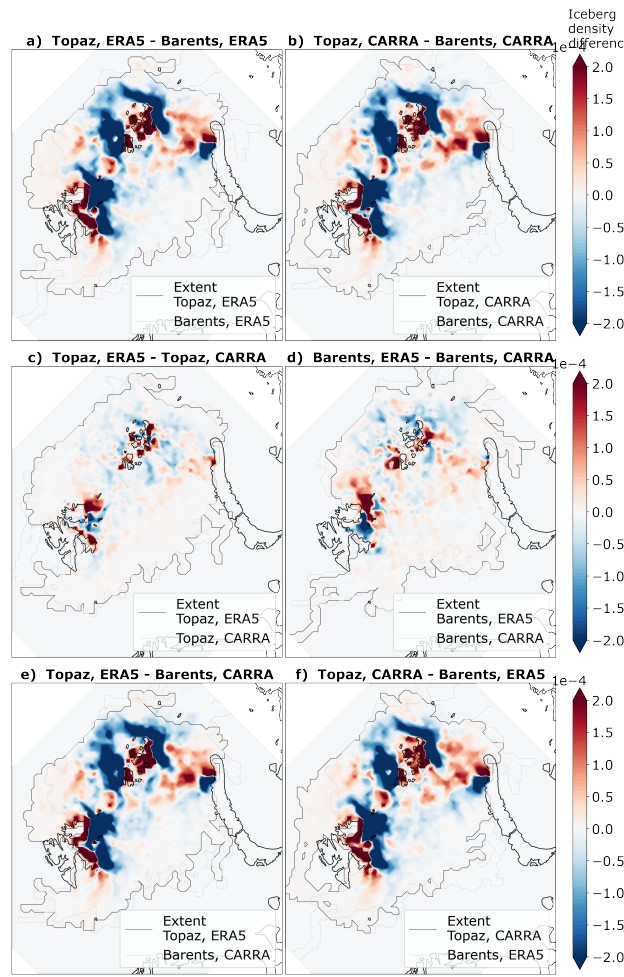

**Figure 5.** Difference of iceberg density (colours) and southernmost extend (lines) in simulations with a,b) varied ocean and sea ice forcing, c,d) varied atmospheric forcing and e,f) variations of both forcing datasets.





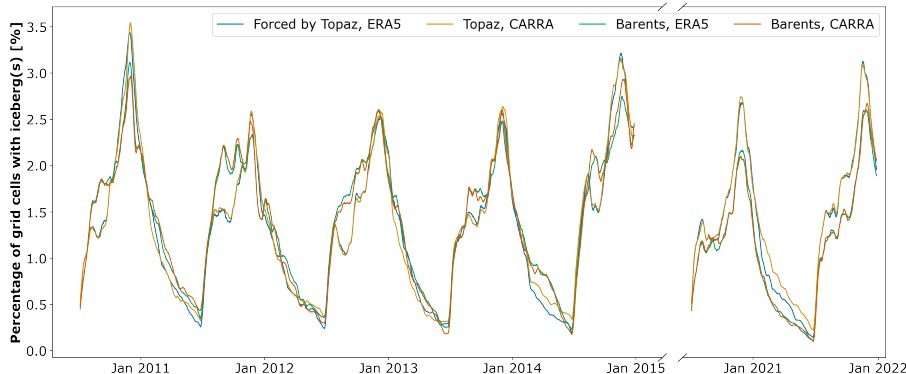

**Figure 6.** Time series of iceberg extension (relative number of grid cells containing icebergs, in % of all grid cells) in the differently forced iceberg simulations from 2010 to 2014 and 2020 to 2021. The extension is relative to the number of analysis grid cells. A 10 days rolling average has been applied.

### 3.6 Example of an iceberg trajectory

The iceberg (referred to as *iceberg 2013-788*) drifted southward from Franz-Josef-Land from autumn 2013 to spring 2014 (Fig.7). The trajectory of iceberg *2013-788* is some of the longest (up to $249\,\text{days}$, $1030\,\text{km}$ effective and $3900\,\text{km}$ track distance) and southernmost trajectories (down to $72°\text{N}$) out of the statistics of $2603 \cdot 7 \cdot 4$ simulated trajectories discussed in this study (Table 3).

Trajectories with different ocean and sea ice forcing deviate significantly in the second half of the drift, drifting into the Central Basin under Barents-2.5 forcing and into the Hopen Trench under Topaz forcing (Fig. 7). The Barents-2.5 trajectory drifted $3°$ further south (Table 3). The Topaz-forced trajectories have a longer drift duration ($+11\,\text{days}$) and distance along the track ($+420\,\text{km}$), but shorter effective drift distance ($-82\,\text{km}$). The trajectories show minor deviations due to varied atmospheric forcing on the large scale.

The environmental conditions in the Barents Sea are shown for selected time steps during the winter 2013-2014 in Fig. 8. The environmental forcing along the trajectory of *iceberg 2013-788* is shown as timeseries in Fig. 9.

Relevant sea ice is restricted to the north and west of Franz-Josef-Land in Topaz at the seeding time in mid October 2013. At the same time, sea ice in Barents-2.5 encloses the archipelago, setting the iceberg in light sea ice conditions (line hatches in Fig. 8). During the winter, the relevant sea ice expands south-east-ward, with light sea ice reaching as far south as Hopen island for Topaz and as far south as Bjørnøya for Barents-2.5 in late April. The cover of heavy sea ice is larger in Barents-2.5 throughout the winter (Fig. 8, point hatches). The sea surface temperature reflects this difference in the spatial distribution



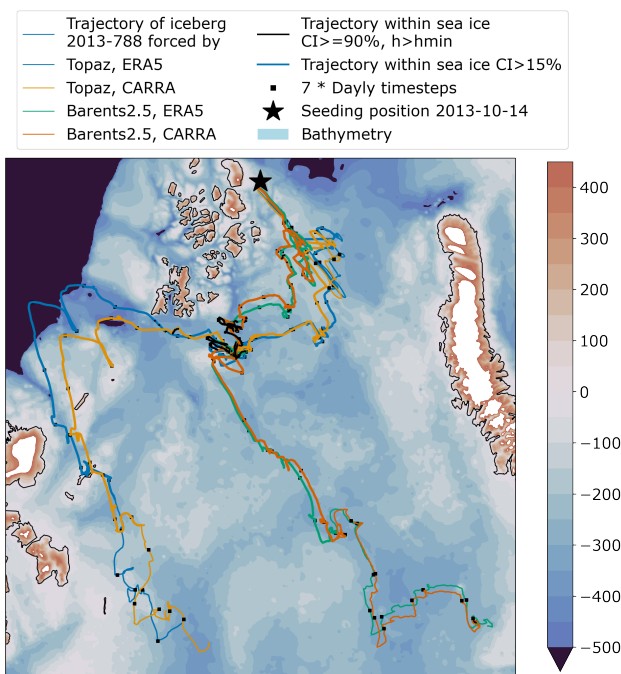

**Figure 7.** Simulated drift of *iceberg 2013-788*, seeded at the 14 October 2013, close to north-eastern Franz-Josef-Land (star). The trajectories correspond to simulations of iceberg drift and deterioration with varied environmental forcing (coloured lines). Along the trajectories, weekly time steps (black squares) and simulation time steps with relevant light sea ice ($CI > 15\%$, thicker lines) and heavy sea ice conditions ($CI \leq 90\%$ and $h_{si} > h_{min}$, black lines) are marked.

**Table 3.** Characteristics of the drift of *iceberg 2013-788*. Iceberg drift duration [days], effective drift distance [km], drift distance along the trajectory [*Track*, km], southern-most latitude [*min lat.* °N] and simulation end date (*Melt date*).

| Forced by | Duration [days] | Effective dist. [km] | dist. Track [km] | min lat. [$^{\circ}N$] | Melt date |
|---|---|---|---|---|---|
| Topaz, ERA5 | 249 | 936 | 3852 | 75.5 | 2014-06-20T06 |
| Topaz, CARRA | 229 | 968 | 3401 | 75.0 | 2014-05-31T04 |
| Barents2.5, ERA5 | 236 | 1038 | 3204 | 72.0 | 2014-06-06T12 |
| Barents2.5, CARRA | 232 | 1031 | 3209 | 72.1 | 2014-06-03T08 |
| Topaz - Barents2.5 | +11 | -82 | +420 | +3.2 | - |
| ERA5 - CARRA | +23 | -12 | +223 | -0.3 | - |

accordingly.

Along the trajectory, the icebergs drift within sea ice (relevant for iceberg drift) $70 - 77\%$ of the simulation days (Table 4, Fig. 9). Thereby Barents-2.5-forced simulations show a larger number of days with relevant sea ice ($+7\%$ of days), average





**Table 4.** Statistics of sea ice along the of the trajectory of *iceberg 2013-788* with relative number of days in conditions with $CI > 15\%$, average sea ice concentration ($CI$), thickness ($h_{si}$), sea surface temperature ($SST$), 10m wind $v_\mathrm{a}$ , sea water surface $v_\mathrm{w}$ and sea ice speed $v_\mathrm{si}$, along the trajectory. The sea ice speed is averaged over time periods with relevant sea ice.

|  | $\% \, CI > 15\%$ | $\varnothing CI \, [\%]$ | $\varnothing h_{si} \, [\mathrm{m}]$ | $\varnothing SST \, [^\circ\mathrm{C}]$ | $\varnothing v_{ai} \, [\mathrm{m\,s{-}1}]$ | $\varnothing v_w \, [\mathrm{m\,s{-}1}]$ | $\varnothing v_a \, [\mathrm{m\,s{-}1}]$ |
|---|---|---|---|---|---|---|---|
| Topaz | 70 | 64 | 0.32 | -1.27 | 0.05 | 0.13 | 6.95 |
| Barents-2.5 | 77 | 74 | 0.42 | -1.26 | 0.06 | 0.16 | 7.32 |
| Topaz- Barents2.5 | -7 | -10 | -0.1 | +0.01 | -0.01 | -0.03 | -0.37 |

**Table 5.** Difference of mean absolute $2-$hourly deterioration rate ($\delta$, $10^4 \, \mathrm{kg \, 2h^{-1}}$) and relative contributions by the deterioration terms ($\delta(term)$, $\%$) in the differently forced trajectories of iceberg 2013-788. The deterioration terms are melt erosion $M_\mathrm{e}$, basal melt $M_\mathrm{fw}$ and buoyant vertical convection $M_\mathrm{v}$

|  | $\delta \, [10^4 \, \mathrm{kg \, 2h^{-1}}]$ | $\delta(M_e) \, [\%]$ | $\delta(M_f w) \, [\%]$ | $\delta(M_v) \, [\%]$ |
|---|---|---|---|---|
| Topaz-Barents2.5 | -0.9 | +5/-5 | -5/-2 | -0.01/-0.2 |

10% larger $CI$ and $0.1\,\mathrm{m}$ larger $h_\mathrm{si}$. The $SST$ along the trajectory is characterised by the present sea ice until April/May 2014
($\approx -2^\circ\mathrm{C}$) and the drift into warmer Atlantic waters (up to $+4^\circ\mathrm{C}$) in the Hopen Trench and the Central Basin (Fig. 7, 9). The $SST$ is in average $0.01^\circ\mathrm{C}$ larger in Topaz-forced trajectories (Table 4).

Wind, sea water and sea ice speed vary across the domain and fluctuate on short temporal scales in the timeseries, especially for Barents-2.5, ERA5 and CARRA (Fig. 9). The speed of the forcing variables is in average larger in Barents-2.5
($+0.01\,\mathrm{m\,s^{-1}}$) for water, $+0.05\,\mathrm{m\,s^{-1}}$ for sea ice, Table 4). Further analysis obtained that the wind speed deviates to a larger degree for varied ocean forcing (position) than atmospheric forcing (wind speed itself, not shown).

The iceberg deterioration rate (individual terms and sum of terms) is small ($20.5 \cdot 10^4 \, \mathrm{kg \, 2h^{-1}}$) during the drift within sea ice and basal melt dominates (Fig. 9, means not shown). When the icebergs start to drift outside of the sea ice, the deterioration
rates increase ($89.1 \cdot 10^4 \, \mathrm{kg \, 2h^{-1}}$) and the contribution by wave erosion dominates. The total contribution by wave erosion is larger in Topaz-forced trajectories ($+5\%$), compared to Barents-2.5 forcing. The opposite is true for the basal melt ($-5\%$). The combined deterioration rate of *iceberg 2013-788* is in average $0.9 \cdot 10^4 \, \mathrm{kg \, 2h^{-1}}$ smaller in Barents-2.5-forced trajectories (Table 5).

We highlight the period between 1 April and 15 May 2014, when the differently forced icebergs drift out of the relevant sea ice, to the east of Svalbard (Topaz-forced trajectories) and in the Central Basin (Barents-2.5-forced trajectories) (Fig. 7). The general environmental situation in April 2013 can be described by the yearly maximum sea ice extent and infusions of warm





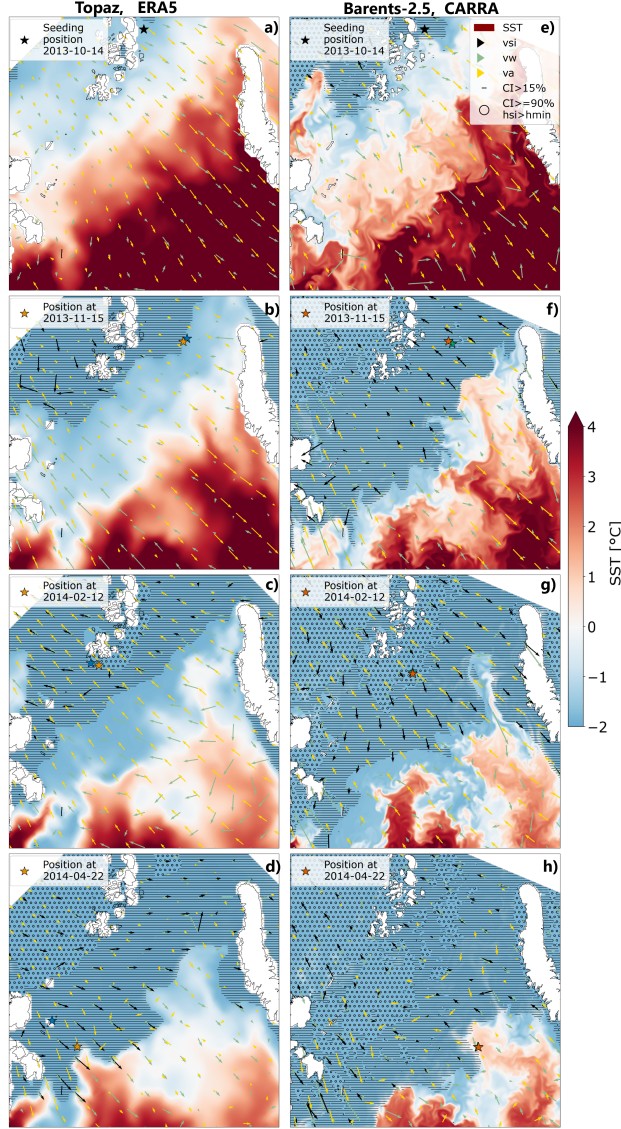

**Figure 8.** Environmental conditions in the Barents Sea at the 14 October 2013 (a,e), 15 November 2013 (b,f), 12 February 2014 (c,g) and 22 April 2014 (d,h). Ocean and sea ice conditions are given by Topaz (a-d) and Barents-2.5 (e-h). The atmospheric conditions are provided by ERA5 (a-d) and CARRA (e-h). Shown variables are sea surface temperature (contour colours), light sea ice ($CI > 15\%$, line hatches), heavy sea ice ($CI \leq 90\%$, $h_{si} > h_{min}$, point hatches), sea ice drift (black arrows), sea surface velocity (green arrows) and 10m-wind (yellow arrows). Note, that the directional data is given for reduced (approx. 100 km resolution) for increased visibility. The respective position of *iceberg 2013-788* (star) is marked for the simulations forced by Topaz and ERA5 (Blue), Topaz and CARRA (orange), Barents-2.5 and ERA5 (green) and Barents-2.5 and CARRA (red).




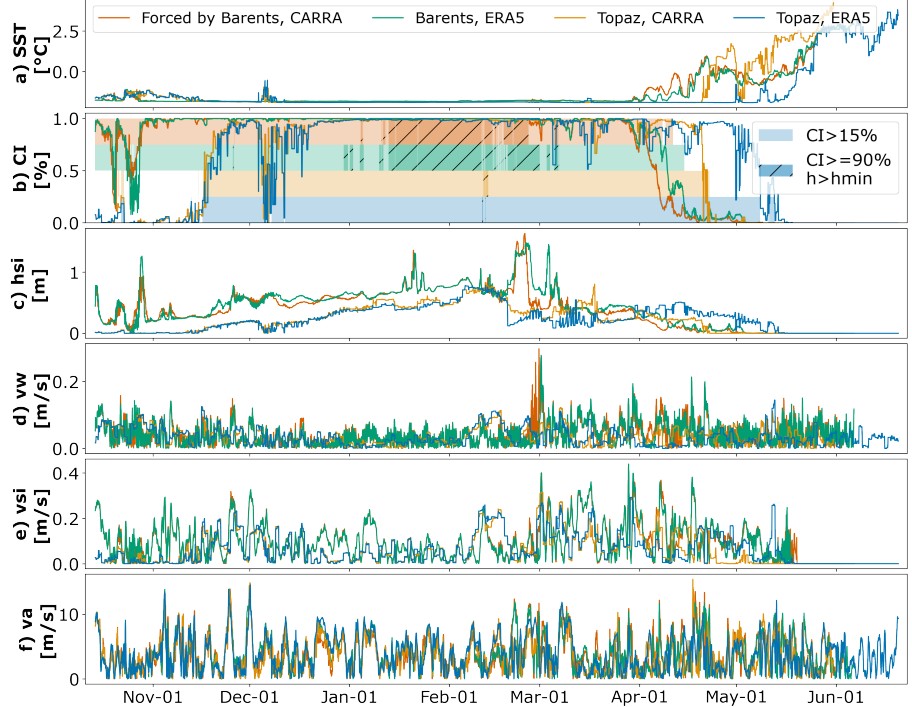

**Figure 9.** Time series of iceberg and forcing characteristics along the trajectory of *iceberg 2013-788*. Environmental forcing along the trajectory with a) sea surface temperature $SST$ [°C], b) sea ice concentration $CI$ and time steps with $CI > 15\%$ (colour) and $CI \leq 90\%$ with $h_{si} > h_{min}$ (colour, hatches), c) sea ice thickness $h_s i$ [m], d) surface water speed $v_w$ [m s$^{-1}$], e) sea ice drift speed $v_{si}$ [m s$^{-1}$], f) 10m wind speed $v_a$ [m s$^{-1}$]. Time series of iceberg deterioration during the drift with g) iceberg mass loss per time step [kg 2h$^{-1}$] and contribution [kg 2h$^{-1}$] by h) wave erosion [$M_e$], i) basal melt [$M_{fw}$] and j) buoyant convection [$M_v$].

Atlantic waters towards the sea ice edge (e.g. along the Hopen Trench and Central Basin) (Fig. 8). Along the trajectories, the icebergs face decreasing sea ice concentration and thickness and increasing $SST$ during their southward drift (Fig. 9). The icebergs also face rapid changes in wind and water speed.

## 4 Discussion

### 4.1 Availability of forcing data

The availability of forcing data influences the simulation results of iceberg drift and deterioration (Bigg et al., 1997; Kubat et al., 2005; Eik, 2009b). Simulations of iceberg drift and deterioration require forcing variables at a given time and position (Sect. 2.4). The availability of gridded forcing variables at this time and position depends on the general availability of the forcing dataset, the forcing data resolution, land-sea-mask, and the iceberg model setup (Herrmannsdörfer et al., 2024b). Herrmannsdörfer et al. (2024b) indicates, that the horizontal resolution and the land-sea-mask of the forcing data may be deciding



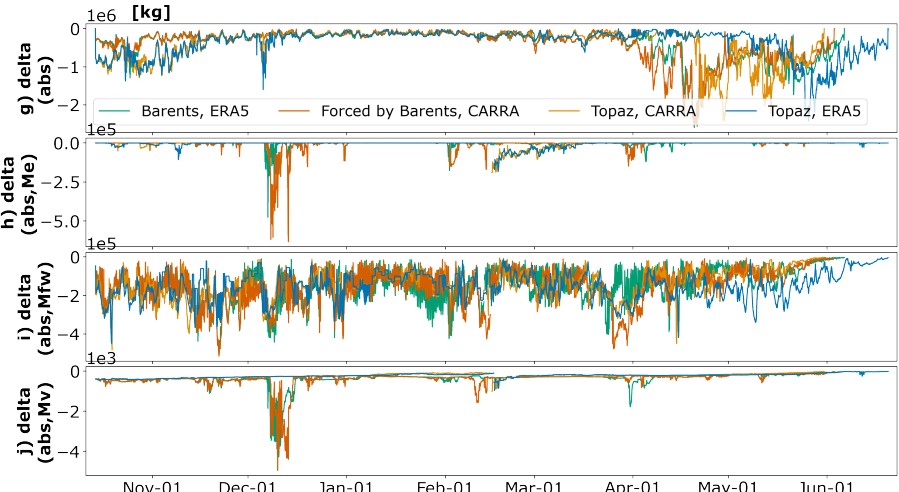

**Figure 9.** (Continuation of Fig. 9

factors for the forcing availability, when icebergs drift close to the coastline. The individual forcing datasets have different bathymetry and coastlines due to their horizontal resolution and grid orientation (rotation relative to latitude-longitude-grid) and type (e.g. curvilinear, regular).


The results of this study confirmed a dependency on the input data sets' horizontal resolution, bathymetry and land-sea-mask. Independent of the forcing dataset and its resolution, the majority of simulated trajectories and time steps have forcing available in the nearest grid cell, and a small minority lack forcing entirely. However, we found that the availability of forcing variables in the nearest grid cell varied with forcing data resolution and land-sea-mask. This small differences matter as they may cause large impact on the iceberg simulations.


We found a dependency on the horizontal forcing resolution and the search radius. In detail, the maximum search radius (in km) within which forcing is available, is smaller for forcing with higher resolution. In addition, high-resolution forcing data allows for small step-wise increase in search radius (acting like a step-wise decrease of resolution). In this example, about $5 \cdot 5$ grid cells ($5 \cdot 2.5\,\mathrm{km}$) of the Barents-2.5 dataset equal one Topaz cell (of about $12.4 \cdot 12.4\mathrm{km}$). Thus, forcing from Barents-2.5 is gathered at a higher resolution than variables from Topaz, even when not available in the nearest forcing cell, and resolution is decreased at a smaller rate. Comparing ERA5 and CARRA, ERA5's coarse resolution, especially in latitudinal direction (due to the regular grid), causes larger search radii. The impact of low horizontal resolution is largest in regions with complex topography and bathymetry.



Further, we find, that availability of low resolution atmospheric variables along the coastlines can be compensated by a more relaxed land-ocean-mask. This was achieved by masking at $25\%/50\%$ grid cell- water surface, according to the supplied land-





mask of CARRA and ERA5. However, depending on the coastal topography, a more relaxed land-ocean-mask may decrease
the representativity of the atmospheric variable for the ocean part of the cell. Note that this cannot be applied for the ocean
variables, as they are produced on a fixed land-sea-mask.

The presented data assimilation approach (Sect. 2.5) and dependency of forcing data availability on the horizontal resolution
are important for a (precise as possible) physical simulation of iceberg close to coastlines. However, both high and low resolu-
tion ocean and sea ice models exhibit large errors in proximity to the coast, such as accumulation of water and ice (Idžanović
et al., 2024). As iceberg simulations bridge the gap that is the lack of a statistically relevant number of observations, the prop-
agation of those inaccuracies into the iceberg simulation must be accepted, and iceberg simulations close to the coast must be
seen in the light of these inaccuracies.

### 4.2 Iceberg drift and deterioration

The theoretical influence of ocean, sea ice and atmosphere forcing on the iceberg drift is contrasting. While larger wind and sea
water surface velocities increase the iceberg velocity and drift distance, they also contribute to the erosional and basal melt, due
to a rougher sea state and higher differential velocity between iceberg and the sea. Positive $SST$ contribute to all deterioration
terms. In the presence of sea ice, $SST$ are around $-1.8°C$, thus the melt is reduced by lower $SST$ and modulated waves.
In sea ice concentrations over $15\%$ ($90\%$), icebergs drift (solely) with the sea ice, setting the drift velocity and direction. We
found significant differences in the iceberg drift and deterioration due to varied forcing.

*Iceberg deterioration*

The relative importance of the deterioration terms along the main iceberg pathways in the Barents Sea reflects the findings
from El-Tahan et al. (1987) and Eik (2009a) in general, with largest contribution by wave erosion. The sensitivity of iceberg
deterioration on sea ice can be seen well for iceberg 2013-788 in April and May 2014, when the icebergs drift out of sea ice
edge and all deterioration terms increase rapidly.

The differences in relative importance in simulations with varied ocean and sea ice forcing highlight the importance of occur-
ring sea ice conditions. As such, iceberg deterioration by wave erosion is decreased in Barents-2.5-forced iceberg simulations
by its more extensive sea ice. In detail, Barents-2.5 provides more simulation time steps and trajectories with sea ice and the sea
ice concentration is higher in the presence of sea ice, in the iceberg pathways (Herrmannsdörfer et al., 2024a). Decreased dete-
rioration (for all terms) in iceberg simulations forced by Barents-2.5 is caused by average lower $SST$ in Barents-2.5 ($-0.41°C$
for iceberg pathways and throughout most of the domain, Herrmannsdörfer et al. (2024a)), due to coupling with excessive
sea ice. Larger basal melt, may be explained by larger water velocities in Barents-2.5, despite larger average Topaz $SST$. As
Barents-2.5 suffers from a excessive representation of sea ice, related too small $SST$ and too large velocities, due to its model
setup, the decreased deterioration may be unrealistic. As the relative contribution of buoyant vertical convection is small, the





differences due to varied forcing are insignificant.

*Iceberg drift duration and distance*

Decreased deterioration rates of Barents-2.5-forced icebergs also favour longer drift duration and drift distance (along the trajectory). Drift duration and Track distance may be further increased by iceberg *looping*, due to tidal forcing.

Slightly larger effective drift distances in Topaz-forced simulations might indicate, that including tidal components and local representation of variables might not alter how far the icebergs drift effectively. One might conclude that the tidal component

is not essential to iceberg statistics in the Barents Sea, however we found that it is essential to simulating individual iceberg trajectories, iceberg occurrence (density) and extent in the domain.

Small differences in iceberg drift duration and distance between ERA5 and CARRA-forced simulations, are dominated by the differences in ocean and sea ice forcing.


Differences in iceberg drift with varied forcing can be derived from different representation of these environmental regimes. As the differences from varied forcing are smaller than than the ones from different regions, we assume that the different forcing data sets could represent the differences between the regionally varying environmental regimes (e.g. different $SST$ and $CI$). This may also be seen e.g. in similar main pathways from spatial densities and spatial extent.

**4.3  Icebergs in the domain**

We found that iceberg distribution and spread in the domain is dependent on the environmental forcing of ocean, sea ice and atmosphere. Spatial density differences may indicate an impact on the simulated iceberg pathways and large regional differences.

*Iceberg density*

We found similar main pathways for all differently forced simulations. Iceberg density is largest in proximity to the iceberg sources, as the average effective drift distance is only around 100 km (Sect.3.3). The density is especially large when icebergs drift (loop) in the same regions for a long time. This is the case for Barents-2.5-forced trajectories, as they life longer due to decreased deterioration rates (Sect. 4.2). As a consequence, average domain and local peak iceberg densities are higher for Barents-2.5-forced simulations.


Varied ocean- and sea ice-forcing cause large regional differences in iceberg density around Svalbard and Franz-Josef-Land. Differences of iceberg density due to varied forcing are as large as the absolute density and thus highly relevant. The iceberg density is larger for Topaz-forced simulations close to the coastlines and larger for Barents-2.5-forced simulations in slightly larger distance to the coastlines. This may be due to larger Barents-2.5 water speeds along the coastlines and slightly larger

Topaz water speeds on the open ocean (Herrmannsdörfer et al., 2024a). It may also be due to different bathymetry, representa-



tion of the coastlines and horizontal resolution in the forcing data sets, that can alter the simulated density by iceberg grounding and the need for neighbour averaging, as described in Sect. 2.4. However, the results along the coastlines need to be viewed in the light of low viability of the forcing data in coastal regions.

Examples of the influence of the ocean and sea ice forcing on the iceberg densities are explained in the following. Higher iceberg densities in Barents-2.5-forced simulations in the northernmost parts of the domain may be explained by the forcing's lower $SST$ and thicker sea ice, that increases the iceberg lifetime and may also trap the iceberg in those regions (Herrmannsdörfer et al., 2024a). Iceberg density differences around northern Novaya Zemlya may be related to large differences in surface water speed and direction and more frequent thick sea ice in the Barents-2.5 forcing (Herrmannsdörfer et al., 2024a) that pro-

hibit a north- and westward drift. Large density differences north of Bjørnøya may evolve from lager forcing differences in $SST$, water velocity and sea ice around the bathymetric feature of the Storfjorden Trough, Spitsbergen Bank and Hopen Trench (Herrmannsdörfer et al., 2024a). In contrast to Topaz, Barents-2.5 has complex spatial and temporal differences in water speed and direction, as it represents more local processes, including a strong tidal component and complex water motion due to the complex bathymetry around the Spitsbergen Bank. For more, the more extensive sea ice cover over the Spitsbergen Bank in

Barents-2.5, might increase the release into open waters of Hopen Trench.

    Varied atmospheric forcing causes large differences in iceberg density on small spatial scales. This may be due to the resolution difference of the forcing datasets, that cause larger CARRA wind speeds in coastal areas and provide (more accurate) forcing between the islands of the Svalbard and Franz-Josef-Land archipelago and in their fjords. Even though, the availability of forcing is assured close to the coast independent of the resolution (Sec. 2.4 and 3.1), the representability in complex topog-

raphy is larger for higher resolution wind. High resolution ocean and sea ice forcing further increases the differences of varying the atmospheric forcing, as it allows the iceberg to drift closer to the coast. The deviation of ERA5 and CARRA over sea ice (Herrmannsdörfer et al., 2024a) are compensated by the decreased sensitivity of the drift to wind forcing. The iceberg density is not impacted by atmospheric forcing on large scales.


    In some regions varied ocean, sea ice, and atmospheric-forcing add up or cancel each other out, however the impact of varied ocean and sea ice forcing is larger. Note, that because of the high resolution of Barents-2.5 and CARRA, icebergs can be forced to drift between islands and into fjords. Note that apparent iceberg occurrences on land are due to accumulating occurrences by a nearest neighbour method on an artificial grid of 25 km horizontal resolution (Sec. 2.4,3.4).


### Iceberg extent

We found that iceberg extent varies spatially and temporally, due to varied environmental forcing. Iceberg spread further in the domain (and in all directions) when forced by Barents-2.5. Especially the southward drift of icebergs is limited by the spatial distribution of $SST$ and sea ice parameters, that are steered by large scale atmospheric patterns, global ocean currents and the

bathymetry. Despite the existing deviations between the forcing data sets, iceberg drifted far south (to approximately 72°N),




independent on the forcing.

The multi-year variability of iceberg extension is inherited from the environmental forcing. Beside environmental parameters with little multi-year variability (e.g. surface ocean velocity), sea ice conditions and wind can vary strongly. However, the multi-year variability of iceberg extension is also influenced by the exact seeding, with size, position and seeding date, which varies within defined parameters every year. The multi-year variability is largely reproduced by varied forcing.

The seasonal iceberg extension is influenced by the environmental conditions, especially by the seasonal cycle of $SST$ and sea ice extent. Sea ice decreases the deterioration rate of the icebergs and may alter the iceberg drift speed, depending on the drift region. Larger temperatures and melt rates outside the sea ice limit the spread in the domain. Sea ice increases the iceberg extension in spring, when icebergs drift far south within the sea ice, as the melt rates within the sea ice are small and sea ice expands far south. After the sea ice retreat in summer, icebergs are exposed to larger deterioration rates, limiting the spread in the domain until the sea ice expands again. Thereby, the onset of sea ice growth in the autumn could be a deciding factor for the iceberg extension later in the year. Seasonal differences in wind speed and direction might contributes as well.

The seasonal cycle of iceberg extent in the Barents Sea sea is reproduced in all differently forced simulations, however with deviations. The variation of iceberg extension with environmental forcing can be seen in the relatively large differences in iceberg extension with varied ocean and sea ice forcing in autumn (August-September) and early winter (November-December) (Fig. 6), that cannot be attributed to the seeding mechanism. Differences with atmospheric forcing are present from December to June.

The seasonal cycle of iceberg extension is also partly due to the seeding mechanisms and partly due to environmental conditions. The seasonal cycle is steered by iceberg seeding between July and November, so that lowest iceberg extension occurs just before the start of seeding in July and the larger extent occurs just after the end of the seeding, in late November. The influence of seeding mechanism on the results must be accounted for in the analysis.

### 4.4 Example

The sensitivity of iceberg simulations to their environmental forcing is illustrated by the example of *iceberg 2013-788*. The example demonstrates how small deviations in the environmental forcing lead to large deviations in the drift trajectory (and further deviation in forcing). This can be seen by diverging trajectories of iceberg 2013-788 under varied ocean and sea ice forcing, despite similar initial conditions. The varied forcing ultimately leads to a different drift duration, drift into different regions, and differently far south, in the Barents Sea. Thus, the varied forcing also causes different potential exposure of structures and ships to icebergs.





The large impact by small changes in the forcing can also be seen by in the time period between 1 April and 15 May, when the iceberg drifts out of the sea ice and into regions with different ocean regimes. There, large changes in the iceberg drift and deterioration are caused by changing environmental forcing (of the same input data set) by small change in iceberg position and different timing. This also highlights the importance of temporal and horizontal resolution of the forcing data. The used forcing data sets showed large regional differences. The use of Barents-2.5 may be beneficial due to its high horizontal and temporal resolution.

The large sensitivity to sea ice forcing can be seen in the large difference in iceberg drift forcing and deterioration upon ejection from the sea ice in spring 2014. It can also be seen by varied forcing causing a variation of exposure to sea ice in autumn, days within sea ice, which in this case causes less days with large deterioration and allows for a drift further south. The sensitivity to sea ice may also derive from its large presence along the iceberg pathways.

The minor impact of atmospheric forcing is shown by the smaller regional deviations of trajectories of iceberg 2013-788 with varied atmospheric input. It must be noted, that the *iceberg 2013-788* is initialised far north with above average size, causing a longer (and further south) drift than the average from the statistics of all $4 \cdot 7 \cdot 2603$ simulated icebergs. As such, the example is not suitable to explain the average differences of iceberg simulations with varied forcing, but serves as illustration of the forcing impact.

## 5  Conclusions

In absence of sufficient iceberg observations in the Barents Sea, numerical simulations of iceberg drift and deterioration are the most reliable source for iceberg statistics. We found that the results of such simulations are sensitive to the input from ocean, sea ice and atmosphere reanalyses or forecasts. The study exhibited both small forcing differences leading to large differences in iceberg trajectories and surprising similarities in the statistics, despite large forcing differences.

Large differences in the assimilated forcing information are caused by spatial and temporal resolution. Their horizontal resolution, bathymetry and coastline influence the availability of forcing information in the data assimilation into the iceberg model, distance of acquisition, representability of forcing information for the iceberg position and ultimately, the iceberg distribution and extent in the domain. This is especially visible along the coastlines. We highlight the importance of the forcing resolution's impact in coastal regions, despite the unreliable forcing information in those regions, due to the lack of other (environmental and iceberg) information in this region.

We found dependencies of the iceberg simulations on all forcing variables, however, largest influence is found for the ocean and sea ice variables. Atmospheric forcing showed minor impact for most aspect for iceberg trajectories and statistics.



Sea ice showed especially large influence on iceberg simulations, e.g. for the forcing along the iceberg trajectory and the iceberg deterioration. Sea ice decreases the iceberg deterioration, thereby increasing the length of the drift duration and trajectory. A longer drift and e.g. looping increase the iceberg density, affecting the distribution in the domain.

515

Although, the effective drift distance is not widely influenced by varied ocean and sea ice forcing, varied ocean and sea ice forcing showed large impact on the occurrence and spread in the domain. Thereby, regional difference in iceberg density due to varied forcing can be high. The dependency of the simulation results on the environmental input highlight the importance of choice in forcing for generating iceberg statistics in the Barents Sea, e.g. for estimating the exposure of structures and ships.

520

Similarities in iceberg simulation results despite large forcing differences may be due to multiple compensating effects by varied forcing. Examples are found in a similar southernmost extent despite large deviations in spatial $SST$, or similar seasonal cycle in iceberg extent despite deviations in onset of sea ice freeze-up and melt. Other similarities may derive from a similar representation of the years and regions in the forcing data sets, despite differences in other aspects of the data sets

525 (e.g. multi-year variability of iceberg extent and similar characteristics for icebergs from the same source). We emphasise the general similarity in the main iceberg pathways, despite varied forcing.

We comment that the study is restricted to the years of 2010-2014 and 2020-2021, the Barents Sea, a selection of four environmental models and the specific setup of the iceberg model. However, the findings may be projectable on the other

530 settings.

*Data availability.* Data from ERA5 and CARRA are retrieved from the Copernicus Climate Data Store (Hersbach et al.; Schyberg et al.). The Arctic Ocean Physics Renanalysis (Topaz) is available in Copernicus Marine (MDS, 2023). The Barents-2.5 forecast and hindcast are stored by MET Norway (MET-Norway, a, b). Geostrophic currents are adopted from Slagstad et al. (1990) and bathymetry is gathered from Jakobsson et al. (2012).

535 **Appendix A: Iceberg model and seeding**

**A1 Iceberg seeding**

2603 icebergs are initialised (*seeded*) with start date, position and length for every simulation year 2010-2014 and 2020-2021. Start dates are drawn randomly at 00 UTC from 1 July to 30 November of the respective year. Start positions are drawn randomly from defined regions around the five main iceberg sources in the Barents Sea (see Fig. A1). Iceberg lengths are

drawn randomly from a generalised extreme value distribution, described in Monteban et al. (2020).

$$f(x|k,\mu,\sigma) = \frac{1}{\sigma}exp(-(1+k\frac{x-\mu}{\sigma})^{-\frac{1}{k}})(1+k\frac{x-\mu}{\sigma})^{-1-\frac{1}{k}} \qquad (A1)$$



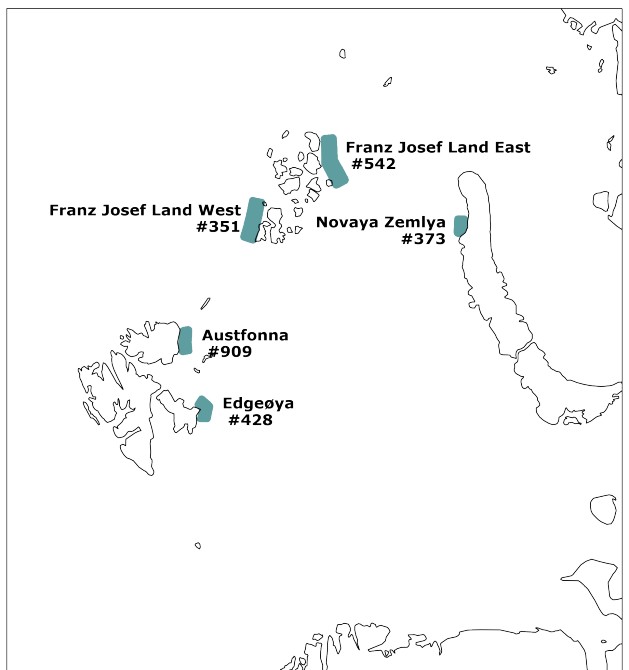

**Figure A1.** Main iceberg sources in the Barents Sea, respective seeding regions (blue) and number of seeded icebergs per simulations year (#).

The distribution is fitted to satellite observations at the main sources, resulting in parameters in Table A1. With the given length $L$, width $W$ and total height (sail plus keel) are calculated by

$$W = 0.7\,L\,exp(-0.000062\,L) \tag{A2}$$

$$H = 0.3\,L\,exp(-0.000062\,L) \tag{A3}$$

Seeding date, position and length are varied for different seeding years and sources, but is reproduced in the differently forced simulations.

## A2   Iceberg model setup and computational routine

The iceberg model components and computational routine are shown in Fig. A2. The iceberg is seeded, then it's velocity $v$ is

updated for every $2-\mathrm{hourly}$ time step $dt$ by calculating the iceberg mass $m$ and the iceberg acceleration $\frac{dv}{dt}$ with the equations of iceberg drift and the environmental input.

$$m = L \cdot W \cdot H \cdot \rho_i \cdot (1 - C_{\mathrm{m}}) \tag{A4}$$

$$\frac{d\boldsymbol{v}}{dt} = \frac{1}{m}[F_{\mathrm{a}} + F_{\mathrm{w}} + F_{\mathrm{c,p}} + F_{\mathrm{si}}] \tag{A5}$$



**Table A1.** Parameters of the generalised extreme value distribution of iceberg length and average iceberg numbers at the main iceberg sources in the Barents Sea.

| Source | Location $\mu$ | Scale $\sigma$ | Shape $k$ | Number $N$ |
|---|---|---|---|---|
| Franz Josef Land West | 44.963 | 14.156 | 0.402 | 351 |
| Franz Josef Land East | 46.480 | 15.636 | 0.252 | 542 |
| Austfonna | 44.501 | 10.668 | 0.118 | 909 |
| Edgeøya | 34.599 | 5.863 | 0.223 | 428 |
| Novaya Zemlya | 39.864 | 12.081 | 0.181 | 373 |

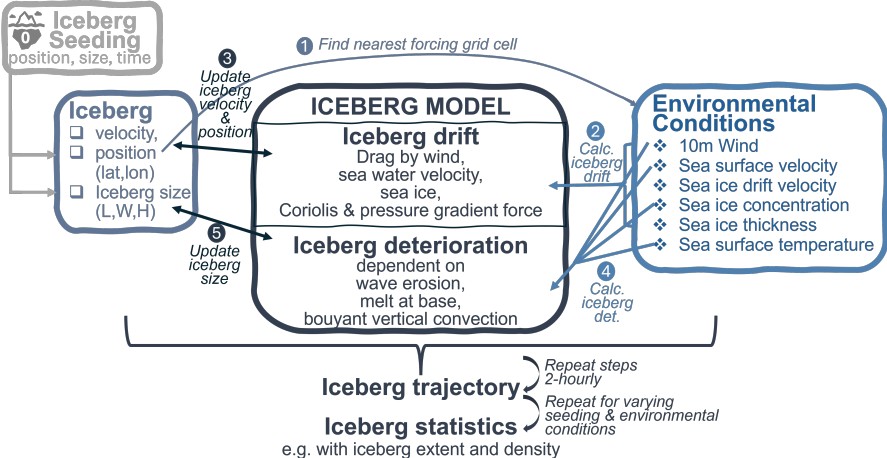

**Figure A2.** Schematic of iceberg model setup with iceberg seeding, drift and deterioration components, environmental forcing, and model output.

Afterwards, the iceberg dimensions $L, W, H$ are updated with the equations of iceberg deterioration and the environmental
input.

$$L = L + (-M_\mathrm{v} - M_\mathrm{e}) \cdot dt \tag{A6}$$

$$W = W + (-M_\mathrm{v} - M_\mathrm{e}) \cdot dt \tag{A7}$$

$$H = H + (-M_\mathrm{b}) \cdot dt \tag{A8}$$

The $2-\mathrm{hourly}$ updates are repeated until the iceberg is melted to the size of a growler ($\mathrm{H} \leq 10\,\mathrm{m}$), leaves the simulation
domain or time period. in order to receive iceberg statistics, this approach is repeated for a large amount trajectories.



## A3 Equations of iceberg drift

Iceberg drift can be expressed by physical iceberg mass $m$, added mass coefficient $C_\mathrm{m}$, iceberg velocity $\boldsymbol{v}$, time $t$, Coriolis force $F_\mathrm{c}$, pressure gradient force $F_\mathrm{p}$, air and water form drag $F_\mathrm{a,w}$, wave radiation stress $F_\mathrm{wd}$ and sea ice forcing $F_\mathrm{si}$ (Savage, 2001).

$$m(1+C_\mathrm{m})\frac{d\boldsymbol{v}}{dt} = F_\mathrm{c} + F_\mathrm{p} + F_\mathrm{a} + F_\mathrm{w} + F_\mathrm{wd} + F_\mathrm{si} \tag{A9}$$

Coriolis and pressure gradient force can be expressed as in Eq. A10, where $u, v$ and $u_\mathrm{geo}, v_\mathrm{geo}$ are the east- and northward components of the iceberg velocity vector $\boldsymbol{v}$ and geostrophic current velocity vector $\boldsymbol{v}_\mathrm{geo}$. Further variables are the Coriolis parameter $f = 2\Omega sin\phi$, Earth's rotation $\Omega = 2\pi \, \mathrm{day}^{-1}$, latitude $\phi$ and the vector normal to the Earth's surface $\boldsymbol{k}$. The geostrophic current is approximated with the geostrophic current $u_\mathrm{geo}, v_\mathrm{geo}$ from Slagstad et al. (1990)

$$F_\mathrm{c,p} = m \cdot [f \cdot v \cdot v_\mathrm{geo}; -f \cdot u \cdot u_\mathrm{geo}] \tag{A10}$$

The form drag due to the surface water current and wind can be written as Eq. A11 and A12, with iceberg drift velocity $\boldsymbol{v}$, near surface water velocity $\boldsymbol{v}_\mathrm{w}$, $10\,m$ wind velocity $\boldsymbol{v}_\mathrm{a}$, water and air density $\rho_\mathrm{w,a}$ and water and air drag coefficient $C_\mathrm{w,a} \approx 1$. The cross section can be described by $A_\mathrm{w} = \frac{\rho_\mathrm{i}}{\rho_\mathrm{w}}\frac{2}{\pi}(L+W)H$ and $A_\mathrm{a} = \frac{\rho_\mathrm{w}-\rho_\mathrm{i}}{\rho_\mathrm{i}}A_\mathrm{w}$ with the iceberg dimensions length $L$, width $W$ and the iceberg sail and keel height $H$. Density effects due to melting and dilution are neglected (Savage, 2001). The influence
by the waves $F_\mathrm{wd}$ is modelled implicitly trough the wind drag coefficient (Monteban et al., 2020).

$$F_\mathrm{w} = \frac{1}{2}\rho_\mathrm{w} C_\mathrm{w} A_\mathrm{w} |(\boldsymbol{v}_\mathrm{w} - \boldsymbol{v})|(\boldsymbol{v}_\mathrm{w} - \boldsymbol{v}) \tag{A11}$$

$$F_\mathrm{a} = \frac{1}{2}\rho_\mathrm{a} C_\mathrm{a} A_\mathrm{a} |(\boldsymbol{v}_\mathrm{a} - \boldsymbol{v})|(\boldsymbol{v}_\mathrm{a} - \boldsymbol{v}) \tag{A12}$$

Sea ice influences the iceberg drift depending on sea ice velocity $\boldsymbol{v}_\mathrm{si}$, sea ice density $\rho_\mathrm{si}$, drag coefficient $C_\mathrm{si}$ and cross section $A_\mathrm{si} = \frac{W+L}{2}h_\mathrm{si}$ (Savage, 2001). The high concentration case is applied under the condition that an ice thickness threshold
$h_\mathrm{si} \geq h_\mathrm{si,min} = \frac{P}{P^* exp(-20(1-CI))}$ with $P = 13000$ and $P^* = 20000$ is fulfilled (Monteban et al., 2020).

$$F_\mathrm{si} = \begin{cases} 0 & \text{if } CI \leq 15\% \\ -(F_\mathrm{c} + F_\mathrm{p} + F_\mathrm{a} + F_\mathrm{w}) + \frac{d\boldsymbol{v}_\mathrm{si}}{dt} & \text{if } CI \geq 90\% \\ \frac{1}{2}\rho_\mathrm{si} C_\mathrm{si} A_\mathrm{si} |(\boldsymbol{v}_\mathrm{si} - \boldsymbol{v})|(\boldsymbol{v}_\mathrm{si} - \boldsymbol{v}) & \text{otherwise} \end{cases} \tag{A13}$$

## A4 Equations of iceberg deterioration

Iceberg deterioration can be described by deterioration due to solar radiation $M_\mathrm{s}$, buoyant vertical convection $M_\mathrm{v}$, forced convection by air and water $M_\mathrm{fw,fa}$, wave erosion $M_\mathrm{e}$ and wave calving $M_\mathrm{cal}$ (Kubat et al., 2007; Eik, 2009a) (Eq. A14).

$$M_\mathrm{total} = M_\mathrm{s} + M_\mathrm{v} + M_\mathrm{fa} + M_\mathrm{fw} + M_\mathrm{e} + M_\mathrm{cal} \tag{A14}$$

The terms contribute to the total deterioration at different rates with highest impact from wave erosion (with calving), forced convection by water, and to a much smaller degree buoyant convection (El-Tahan et al., 1987; Savage, 2001; Kubat et al.,



**Table A2.** Coefficients of iceberg drift and deterioration.

|            | Description               | Value | Reference                                    |
| ---------- | ------------------------- | ----- | -------------------------------------------- |
| $C_\mathrm{m}$  | Added mass coefficient    | 0     | Keghouche et al. (2009); Monteban et al. (2020) |
| $C_\mathrm{w}$  | Water drag coefficient    | 0.25  | Keghouche et al. (2009)                      |
| $C_\mathrm{a}$  | Air drag coefficient      | 0.7   | Monteban et al. (2020)                       |
| $C_\mathrm{si}$ | Sea ice drag coefficient  | 1.0   | Eik (2009b)                                  |

**Table A3.** Physical parameters of ocean, atmosphere and sea ice for the simulation of iceberg drift and deterioration.

|             | Description        | Value                         | Reference              |
| ----------- | ------------------ | ----------------------------- | ---------------------- |
| $\rho_\mathrm{w}$  | Water density      | $1027\,\mathrm{kg\,m^{-3}}$   | -                      |
| $\rho_\mathrm{a}$  | Air density        | $1.225\,\mathrm{kg\,m^{-3}}$  | -                      |
| $\rho_\mathrm{si}$ | Sea ice density    | $900\,\mathrm{kg\,m^{-3}}$    | -                      |
| $\rho_\mathrm{i}$  | Iceberg density    | $850\,\mathrm{kg\,m^{-3}}$    | Monteban et al. (2020) |
| $T_\mathrm{i}$     | Iceberg temperature | $-4^\circ C$                 | Wagner et al. (2017)   |

2007; Eik, 2009a). In this study, the effect of solar radiation and forced convection by wind is neglected. Due to its complexity, calving is not explicitly described. The erosional melt $M_\mathrm{e}$ due to waves is described by sea surface temperature $SST$, sea ice

concentration $CI$ and sea state $Ss = \frac{3}{2}|V_\mathrm{a} - V_\mathrm{w}|^{0.5} + 0.1|V_\mathrm{a} - V_\mathrm{w}|$ with total wind and current speed $V_\mathrm{a,w}$ (Eq. A15).

$$M_\mathrm{e} = \frac{(\frac{1}{6}[SST+2])Ss(0.5[1+cos(CI^3\pi)])}{24\cdot 3600} \tag{A15}$$

The melt due to buoyant vertical convection $M_\mathrm{v}$ is given by the freezing point temperature $t_\mathrm{fp} = t_\mathrm{fs}\cdot exp(-0.19\cdot[SST - t_\mathrm{fs}])$, sea water freezing temperature $t_\mathrm{fs} = -0.036 - 0.0499\cdot Sal - 0.000112\cdot(Sal^2)$ and Salinity $Sal = 34.8$ (Eq. A16).

$$M_\mathrm{v} = 8.8\cdot 10^{-8}[SST - t_\mathrm{fp}] + 1.5\cdot 10^{-8}[SST - t_\mathrm{fp}]^2 \tag{A16}$$

The forced convection by water $M_\mathrm{fw}$ or turbulent basal melt is calculated by the East/North component of the iceberg and water drift $u$, $v$ and $u_\mathrm{w}$, $v_\mathrm{w}$, the iceberg length $L$ and the ice temperature close to the water interface $T_\mathrm{i}$ of $-4^\circ C$ (Eq. A17).

$$M_\mathrm{fw} = 6.7\cdot 10^{-6}\sqrt{(u - u_\mathrm{w})^2 + (v - v_\mathrm{w})^2}^{0.8}\cdot(SST - T_\mathrm{i})\cdot L^{-0.2} \tag{A17}$$

## A5   Model parameters

Table A2 and A3 show the parameters used in the iceberg simulations.

*Author contributions.* Data pre-processing, model adaptions, simulations, statistical analysis and original draft of manuscript: LH. Supervision during all stages of the study and review of the manuscript: RKL, KVH.



*Competing interests.* The authors declare that they have no conflict of interest.

*Acknowledgements.* The authors wish to acknowledge the support from the Research Council of Norway through the RareIce project (326834) and the support from all RareIce partners. The authors also wish to acknowledge Dennis Monteban, for supporting the under-standing of the iceberg model.


The ERA5 (Hersbach et al.) and CARRA data (Schyberg et al.) were downloaded from the Copernicus Climate Change Service (2023). The results contain modified Copernicus Climate Change Service information 2023. Neither the European Commission nor ECMWF is responsible for any use that may be made of the Copernicus information or data it contains. This study has been conducted using E.U. Copernicus Marine Service Information, (MDS, 2023).




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
