# Peer review of "Sensitivity of iceberg drift and deterioration simulations to input data from different ocean, sea ice and atmosphere models in the Barents Sea."

_EGUsphere, 2024_

## Referee Comment (RC1)

**Review of " Sensitivity of iceberg drift and deterioration simulations to input data from different ocean, sea ice and atmosphere models in the Barents Sea (Part II)"**

**General Comments**

This paper investigates the sensitivity of iceberg drift and deterioration simulations to varying input data from different ocean, sea ice, and atmospheric models in the Barents Sea. It provides a detailed numerical experiment using four combinations of environmental forcing data (Topaz, Barents-2.5, ERA5, and CARRA). The study emphasizes that iceberg drift and deterioration are particularly sensitive to the choice of ocean and sea ice forcing data (Topaz or Barents-2.5). While the combination of statistical analysis and individual case studies, such as the trajectory of iceberg 2013-788, adds depth to the study, I find some critical gaps and ambiguities in the paper.

Firstly, the motivation for this study is unclear. While the authors explicitly demonstrate differences in iceberg deterioration under various forcing scenarios, the key takeaway remains vague. What actionable insights can we derive from these results? Which combination provides the most reliable or "best" estimate of iceberg drift? Without addressing these questions, the practical value of this study seems limited.

Secondly, the temporal scope of the analysis raises concerns. The authors focus on limited time windows (2010–2014, 2020–2021), and yet they find that atmospheric forcing has only minor effects. Why not extend the analysis to the longer overlapping period (2010–2022) available for Topaz, Barents-2.5, and ERA5? This would allow for a more robust assessment of iceberg occurrences and their trends over the full 12 years. The decision to restrict the study to only seven years until the end is not adequately justified.

Thirdly, the use of the nearest-neighbor scheme for environmental forcing is perplexing. The authors spend considerable text explaining the process, but why was this method chosen over more commonly used interpolation techniques such as linear or inverse distance weighting? How can the errors introduced by this nearest-neighbor approach, especially near coastlines, be quantified? This choice introduces uncertainty, which is not thoroughly analyzed.

Finally, Table 1 lacks clarity regarding the spatial resolution used in each test. In a sensitivity analysis, it is critical to change one parameter (e.g., spatial resolution, ocean/ice forcing, or atmospheric forcing) while keeping the others constant. The reference setting for the experiments remains unclear. And why keeps the original resolution instead of interpolating into different resolution?

In summary, the paper has potential but requires major revisions to address these concerns before it can be considered for publication.

**Specific Comments:**

1. Figures: Please review all figures to ensure the color legends are consistent and correctly labeled.
2. Line 59: I am not sure that I totally get the sentence, since you must list the rationale for selecting datasets (e.g., Topaz and CARRA) for specific variables like regional wind and their respective resolutions needs to be explicitly stated.
3. Line 99: Why were 2-hourly time steps chosen? Could hourly time steps provide a more precise representation of iceberg drift? Please clarify if 2-hourly intervals are sufficient.
4. Line 105-106: The reasoning for using the nearest-neighbor scheme is unclear. This method is known to be less accurate, especially near coastlines. Why was it chosen over more sophisticated interpolation methods? How do you quantify the associated errors?

5. Figure 7, The influence of bathymetry on Topaz+ERA5 and Topaz+CARRA combinations is intriguing. Could you elaborate on whether this arises from assimilation processes or the original resolution of the models? How can you ensure the observed differences are due to the coarse resolution of ERA5? Are these differences consistently observed in other trajectories over the same regions?
6. Line 433: I am not sure I can get the logics in the discussion on compensatory effects (e.g., similarities in iceberg pathways despite large differences in forcing). Did you compare products with the same forcing but at different resolutions? A clearer explanation is needed.
7. I am not sure if I am convined by just one case in Sec 3.6, did you find some common features based on all extreme case? Relying on a single case study is insufficient to draw broader conclusions. It would be better to incorporate additional deterioration processes into the model to better capture iceberg dynamics under extreme conditions.
8. Mechnism analysis: I would also love to see more mechanism anslysis about wave erosion, basal melt, and buoyant convection since it exactly shows how iceberg react to different forcings. This would clarify how icebergs respond to different forcing conditions. Additionally, could you explore ways to incorporate the uncertainties related to solar radiation, calving, and wave interactions, which are currently simplified in the model?

---

## Author Comment (AC1)

**Author's response to anonymous review of " Sensitivity of iceberg drift and deterioration simulations to input data from different ocean, sea ice and atmosphere models in the Barents Sea (Part II)"**

General Comments

This paper investigates the sensitivity of iceberg drift and deterioration simulations to varying input data from different ocean, sea ice, and atmospheric models in the Barents Sea. It provides a detailed numerical experiment using four combinations of environmental forcing data (Topaz, Barents-2.5, ERA5, and CARRA). The study emphasizes that iceberg drift and deterioration are particularly sensitive to the choice of ocean and sea ice forcing data (Topaz or Barents-2.5). While the combination of statistical analysis and individual case studies, such as the trajectory of iceberg 2013-788, adds depth to the study, I find some critical gaps and ambiguities in the paper.

[1] Thanks so much for your well laid-out summary. The authors' answers to your general and specific comments are given below. In addition, we provide suggestions on how the manuscript may be changed to satisfy your comments. We include a revised version of the conclusions to show an example of possible improvements to the manuscript with the help of the reviewer's comments. The rest of the manuscript will be revised at a later stage of the review process.

*Revised conclusion:*

*In absence of sufficient iceberg observations in the Barents Sea, numerical simulations of iceberg drift and deterioration are the most reliable source of data for iceberg statistics. We forced a large number of iceberg trajectories with varying forcing conditions. We quantitatively confirm and novelly describe how the results of such simulations are sensitive to the input from ocean, sea ice and atmosphere reanalyses or forecasts. The findings may be used in the choice of forcing input and the critical analysis of iceberg model simulations.*

*We confirm that all forcing variables impact the iceberg simulations and quantify how the impact of atmospheric forcing is small compared to the impact of ocean and sea ice variables. Sea ice showed especially large influence on iceberg simulations, e.g. for the statistics of iceberg deterioration or the forcing along an exemplary iceberg trajectory. Sea ice decreases the iceberg deterioration, thereby increasing the length of the drift duration and trajectory. An extended drift duration and e.g. tidal looping increase the iceberg density, affecting the distribution in the domain. Thereby, difference in regional iceberg density and the spread of icebergs in the domain due to varied ocean and sea ice forcing can be high. The dependency of the simulation results on the environmental input data highlights the significant of the choice of such input data for the generation of iceberg statistics in the Barents Sea, e.g. for estimating the probability of iceberg exposure by structures and ships.*

*Although the forcing data has a strong influence on the results of iceberg simulations, it is possible that different combinations of input data yield similar overall results. Similarities in iceberg simulation results despite large forcing differences may be due to multiple compensating effects by varied forcing. Examples are found in a similar effective drift distance and southernmost extent despite large deviations in spatial SST, or similar seasonal cycle in iceberg extent despite deviations in onset of sea ice freeze-up and melt. Other similarities may derive from a similar representation of the year's conditions and regions in the forcing data sets, despite differences in other aspects of the data sets (e.g. multi-year variability of iceberg extent and similar characteristics for icebergs from the same source). We emphasise the general similarity in the main iceberg pathways, despite varied forcing. Those similarities may indicate*

*that the choice of forcing input has limited impact on some applications, e.g. estimating climatology of southernmost iceberg spread in the Barents Sea.*

*We also showed how the assimilation of the forcing data into the iceberg model can be sensitive to the spatial and temporal resolution, bathymetry and coastline of the input data. In the setup of the iceberg model used in this study, the input data's characteristics steer from how far forcing data is assimilated and how representative it is. It influences the simulated iceberg distribution and extent in the domain and is especially visible along the coastlines. We highlight the importance of the forcing resolution's in coastal regions, despite the unreliable forcing information in those regions, due to the lack of other (environmental and iceberg) information. Although the conducted analysis is specific to the model setup, similar effects are expected to occur using other setups.*

*We highlight that the study is restricted to the years of 2010-2014 and 2020-2021, the Barents Sea, a selection of four environmental models and the specific setup of the iceberg model. This is due to temporal and spatial availability of the forcing data and the focus on the impact of varied forcing in the study. We also emphasise that we cannot provide clear suggestions on the best choice of forcing data in iceberg simulations, due to the diverse characteristics of the input data and the faceted impact on the simulations. However, the findings may be projectable on other settings and will facilitate the educated choice in forcing data.*

*Firstly, the motivation for this study is unclear. While the authors explicitly demonstrate differences in iceberg deterioration under various forcing scenarios, the key takeaway remains vague. What actionable insights can we derive from these results? Which combination provides the most reliable or "best" estimate of iceberg drift? Without addressing these questions, the practical value of this study seems limited.*

[2] We attribute this confusion to the lack of sufficient clarifications in the manuscript which should certainly be improved. In the following, we argue that our study is novel and has a practical value, in more detail:

- The motivation of this study is that scientists and engineers in the field rely heavily on statistics of icebergs derived from numerical simulations of icebergs drift and deterioration. This is necessary due to the scarcity of iceberg observations, especially in the Barents Sea. As discovered earlier (Kubat et al. (2005, 2007); Eik (2009b, a); Keghouche et al. (2009, 2010)), the environmental forcing clearly steers the simulations results. Following, we need to estimate the impact of the environmental input on the simulation results. To the knowledge of the authors, no in-depth study of this nature is available, especially not with nowadays state-of-the-art environmental data. This suggests practical value and a high importance of the study for the field. Thereby, the selected environmental data reflects a small range of state-of-the-art environmental models that are likely chosen for simulating icebergs in the study domain.
- We understand the wish for clear outcomes and the rating of a "best estimate". The key-takeaway of this study is that we quantitatively confirm and describe the sensitivity of iceberg simulations to the selected environmental forcing. However, this study restrains from rating best or most reliable estimates for the following reasons:
  - We chose to restrain from varying the model settings to keep the focus on the effect of varying the environmental input data on the simulation results. Varying the model setting would have generated additional sample data that would easily clutter this study. Recall, the goal of this study is not to develop, verify and

validate a model, it is to study the effect of varying the environmental input data on the model results. Therefore, the choice of the model is trivial herein.

- o It is not easy to make a straightforward and simple suggestion about the suitability of the data for iceberg simulations, due to the fact that the environmental models perform better/worse in different regions and times. Although, this is inconvenient for the reader, different ocean, sea ice and atmospheric forcing variables may be differently suitable in different regions, during different time periods, for different variables, iceberg model settings and study goals.

  For example, Topaz and Barents-2.5 show clear advantages and disadvantages and are more/less accurate in different aspects. Topaz is coarser and neglects tides. Barents-2.5 aims at being more accurate with higher temporal and spatial resolution. However, it showed clear overestimation of sea ice. Other variables, such as ocean currents, are highly uncertain in both cases.

  Iceberg models treat the forcing data differently and are differently tuned, which may cause better results with one or the other forcing.

- o The suitability of the forcing also depends on the simulation goal (e.g. statistics of iceberg occurrence or individual trajectories) and which processes it should represent (e.g. tides).

- o Instead of judging the suitability of the forcing input, we tried to outline which specific characteristics of an individual model (e.g. tides in Barents-2.5) cause which impact in iceberg simulations (e.g. spatial distribution). Other planned studies will concern themselves with rating the performance of the iceberg simulations under varied forcing by comparing to iceberg drift observations.

- Based on the reviewer's comments, we suggest following changes in the manuscript.
  - o We will highlight the motivation and importance of the study even more.
  - o Further, we will outline the settings, scope and limitations more clearly.

*Secondly, the temporal scope of the analysis raises concerns. The authors focus on limited time windows (2010–2014, 2020–2021), and yet they find that atmospheric forcing has only minor effects. Why not extend the analysis to the longer overlapping period (2010–2022) available for Topaz, Barents-2.5, and ERA5? This would allow for a more robust assessment of iceberg occurrences and their trends over the full 12 years. The decision to restrict the study to only seven years until the end is not adequately justified.*
[3]

- The choice of limiting the analysis on the years 2010-2014 and 2020-2022 was entirely based on the limited availability of the Barents-2.5 hindcast product, at the time the study was conducted. (CARRA was available for the full time period all the time.) We wanted to include the largest number of years possible, causing the un-intuitive break between 2014 and 2020. Although large variability exists from year to year, we also believe that these two periods might at least characterise the different decades with different sea ice regimes to some point.
  - We suggest to describe the reasoning behind the temporal limitations more clearly in the manuscript.
    - o "The simulations are performed for the years 2010-2014 and 2020-2021 (7 years), which are the only years all forcing datasets were available at the time the simulations were performed. Future studies may concern themselves with analysing the newly available, extended time period."

*Thirdly, the use of the nearest-neighbor scheme for environmental forcing is perplexing. The authors spend considerable text explaining the process, but why was this method chosen over more commonly used interpolation techniques such as linear or inverse distance weighting? How can the errors introduced by this nearest-neighbor approach, especially near coastlines, be quantified? This choice introduces uncertainty, which is not thoroughly analyzed.*
[4]

- Thank you for highlighting the downside of this set-up. We fully agree on the availability of more accurate weighting and interpolation approaches. However, as mentioned above, this study does not aim to analyse or improve the iceberg model setup. Instead, the iceberg model was inherited from Monteban (2020) with most of the settings and kept constant during the experiment. Amongst others, this includes assimilating of the environmental forcing from the nearest grid cell and 2-hourly time steps. As this study does not concern itself with optimising the model setup, quantifying the error induced by nearest neighbour approach is also not part of this study.
- The data assimilation method was described thoroughly in the manuscript on-hand, as in-accurate implementation can lead to physically wrong results. The authors therefore tried to assure the reproducibility of the study.
    - We therefore suggest to mention the existence of more accurate assimilation methods. We also suggest to take away the focus from the data assimilation method, to not confuse the reader about the goals and scope of the work.
    - The description of the assimilation method explains the data pre-processing methods in part I and is the base for the analysis of forcing data availability (Section 4.1). Section 4.1 provided very interesting results, that we wanted to include. It may be transferable on more accurate assimilation techniques, as they all have the problem of missing data e.g. along the coastlines.
    - We know see that Section 4.1 is very specific to the used iceberg model setup and that it may not be easy to transfer the knowledge on other assimilation techniques. If the reviewer or editor think that it would enhance the readability of the paper, we would be able to exclude Section 4.1 and the detailed description of the data assimilation/resampling methods from the manuscript of Part I and II.

*Finally, Table 1 lacks clarity regarding the spatial resolution used in each test.*
[5]

- All environmental data has been used at original spatial resolution. The iceberg model itself is grid-less.
- We suggest to add the spatial resolution (and variables, see answer [10]) of the environmental datasets to Table 1:

| *Objective* | *Ocean & sea ice Forcing \*, spatial, temp. resolution* | *Atmospheric Forcing \*\*, spatial, temp. resolution* |
|---|---|---|
| *Reference* | *Topaz, 12.4 km, daily* | *ERA5, 31 km, hourly* |
| *Regional wind* | *Topaz, 12.4 km, daily* | *CARRA, 2.5 km, 3-hourly* |
| *Regional ocean & sea ice* | *Barents-2.5, 2.5 km, hourly* | *ERA5, 31 km, hourly* |
| *High resolution, fully regional* | *Barents-2.5, 2.5 km, hourly* | *CARRA, 2.5 k, 3-hourly* |

*\* Ocean & sea ice variables: sea surface velocity (vw), surface temperature (SST), sea ice concentration (CI), thickness (hsi) and drift velocity (vsi)*

*\*\* Atmospheric variables: 10m wind (va)*

*In a sensitivity analysis, it is critical to change one parameter (e.g., spatial resolution, ocean/ice forcing, or atmospheric forcing) while keeping the others constant.*

[6]
- We agree that this is the set up of a full sensibility analysis. In this study, we did not perform  full sensitivity analysis, as changing out individual variables or changing resolutions is not a typical use case (see Section). In typical use case, a number of variables will be adapted from one environmental dataset and a number of variables will be taken from other datasets at their respective original resolution and with the respective physical description. We tried to follow the most-likely use case with common environmental models for atmosphere, and ocean/sea ice separately, at their original resolution and interpretation of the variables. The aim of this approach is to show the impact of the forcing data, that may occur in any iceberg simulation as realistic as possible.
- Another reason for this approach was consistency within the forcing input. Changing individual variables and their resolution may give inconsistent forcing (e.g. between sea surface temperature and sea ice concentration).
- We suggest to describe this approach more clearly in the manuscript to avoid confusions with a full sensitivity analysis.
  - Line 59-60: "*We did not conduct a full sensitivity, analysis exchanging every variable individually, to avoid physically inconsistent forcing (of e.g. SST and CI) and to resemble a probable use case as closely as possible.*"
  - *Add at line 72: "All environmental data is used at original spatial resolution, which is indicated in Table 1".*

*The reference setting for the experiments remains unclear.*

[7] We are unsure if this question relates to the setting of the reference experiment (Topaz,ERA5) or the fixed settings  of the iceberg model.
- The settings of the iceberg model are largely adapted from Monteban (2020), that did the latest contribution to the underlaying iceberg model. Those are described in Section 2.4 and the Appendix.
- The setting of the reference experiment is recreated from Monteban (2020), using ocean and sea ice variables from the Arctic Oceans Physics Reanalsyis (Topaz4) and wind from ERA5. Due to the common use of those models and their good spatial and temporal availability, we consider this a valid choice. The choice in iceberg seeding is based on observed size distributions and numbers of iceberg, as presented by Monteban (2020).
- We can try to describe those settings more clearly, if wanted.

*And why keeps the original resolution instead of interpolating into different resolution?*

[8] The authors don't see reasons to change the resolution of the input data in this study. The reasoning for the approach of using original resolution is described in answer [6].

*In summary, the paper has potential but requires major revisions to address these concerns before it can be considered for publication.*

*Specific Comments:*

*1. Figures: Please review all figures to ensure the color legends are consistent and correctly labeled.*

[9] Thank you for pointing this out. We will assure this in the next review stage.

*2. Line 59: I am not sure that I totally get the sentence, since you must list the rationale for selecting datasets (e.g., Topaz and CARRA) for specific variables like regional wind*

*and their respective resolutions needs to be explicitly stated.*

[10] Thank for pointing this out. We will try to describe this more clearly. We suggest to add the variables and resolution to Table 1 (see in answer [5]). Further We suggest to change the text of Section 2.1, e.g.

- Line 57: *The forcing combinations represent a reference case with global forcing (Topaz,12.4 km and ERA5, 31 km) and a high-resolution, regional simulation (with Barents-2.5 and CARRA, both 2.5 km).*
- Line 58-59: *The combinations Topaz and CARRA (12.4 km, 2.5 km) and Barents-2.5 and ERA5 (2.5 km, 31 km) serve to estimate the individual influence of ocean, sea ice and atmosphere forcing on the simulations results.*

The reasoning for not conducting a full sensitivity analysis is explained in answer [6] and the line 59-60 will be improved as described.

*3. Line 99: Why were 2-hourly time steps chosen? Could hourly time steps provide a more precise representation of iceberg drift? Please clarify if 2-hourly intervals are sufficient.*

[11] The 2-hourly time steps derive from the model setup, inherited from Monteban (2020). In this set-up, ocean and sea ice variables are provided at daily frequency (Topaz) and ERA5 data is available hourly, read 2-hourly, but does not influence the simulations to a large degree. Thus 2-hourly time steps are feasible for the reference case. CARRA is only available 3-hourly. The Barents-2.5 data now offers the opportunity of hourly ocean and sea ice forcing. Thus, hourly simulation time steps (and assimilation of the forcing data) may have increased accuracy of the representation of the fast processes. However, as described in answer [2], the goal of the study is not to analyse the impact of the model settings.

- We suggest to describe the scope and limitations of the study more explicitly, as mentioned in answers [2,4].

*4. Line 105-106: The reasoning for using the nearest-neighbor scheme is unclear. This method is known to be less accurate, especially near coastlines. Why was it chosen over more sophisticated interpolation methods? How do you quantify the associated errors?*

[12] See answer [4].

*5. Figure 7, The influence of bathymetry on Topaz+ERA5 and Topaz+CARRA combinations is intriguing. Could you elaborate on whether this arises from assimilation processes or the original resolution of the models? How can you ensure the observed differences are due to the coarse resolution of ERA5? Are these differences consistently observed in other trajectories over the same regions?*

[13]

- The observed differences of the trajectories 2013-788 forced by Topaz+ERA5 and Topaz+CARRA are likely a combinations of effects from horizontal resolution (and topography near coast), assimilation technique, representation of wind over sea ice and growing difference after small initial perturbation.
- The bathymetry/topography should not cause any difference in between the simulated trajectories of Topaz+ERA5 and Topaz+CARRA in distance (>31 km) to the coast. A very small amount of simulated time steps is close to the coastline (e.g. around Kong Karls Land), where the large differences in the wind data exist.
- The assimilation technique is similar for ERA5 and CARRA. As a direct effect of spatial resolution, coarser ERA5 resolution cause larger maximum search radii, potentially making the assimilated data less representative for the iceberg position.
- ERA5 and CARRA are very similar over open ocean (e.g. Køltzow, 2022). Part 1 of this study revealed a systematic difference of ERA5 and CARRA over sea ice, which may

cause the differences in the trajectories in Figure 7, as sea ice is shown by Topaz from late November 2013 to late April 2014.

- We should also keep in mind, that small initial difference in (wind-) forcing may cause small differences in iceberg position, which can turn into large difference of the trajectory in the end.
- Thus, the differently resolved wind input causes difference in the simulated trajectory 2013-788
    - ..due to differently resolved topography along the coastlines.
    - ..due to larger search radius during the assimilation process.

  But differences are also caused by different physical representation of the variable (e.g. over sea ice) and growing difference,
- Visible differences of the simulations forced by Topaz+ERA5 and Topaz+CARRA are also found in the statistics of spatial density (Figure 5). Figure 5 shows how often, how long and how many icebergs reside in different regions of the Barents Sea. Figure 5c varies the atmospheric forcing under Topaz forcing and shows how ERA5 and CARRA cause different density, also along the trajectory of iceberg 2013-788. The same is observed for varied atmospheric under Barents-2.5. Thus, the results of the statistical analysis support the mentioned differences observed for iceberg 2013-788.
- I suggest to add to the (or describe more clearly in the ) discussion of the trajectory:
    - Difference between trajectory is likely a combination of effects, also describing the effects.
    - Connect differences in trajectory 2013-788 with results from density maps.

*6. Line 433: I am not sure I can get the logics in the discussion on compensatory effects (e.g., similarities in iceberg pathways despite large differences in forcing). Did you compare products with the same forcing but at different resolutions? A clearer explanation is needed.*

[14] Thank you for pointing out the lack of clarity concerning the compensatory effects.

- We only worked with the original resolution of the products, however the input comes at different (original) resolutions.
- The mentioned compensatory effects (e.g., similarities in iceberg pathways despite large differences in forcing) derive from the interplay of effects, such as those from provided resolution, but also physical representation of the variables.
- Note, why this study does not contain a full sensitivity analysis, in answer [6].
- We suggest that to review the discussion concerning the mentioned points.

*7. I am not sure if I am convined by just one case in Sec 3.6, did you find some common features based on all extreme case? Relying on a single case study is insufficient to draw broader conclusions. It would be better to incorporate additional deterioration processes into the model to better capture iceberg dynamics under extreme conditions.*

[15]

- We agree that broad conclusions cannot be drawn from one example. The example of the trajectory 2013-788 is meant to illustrate the conclusions from the statistical analysis, rather than to draw new conclusions. One example for this is described in answer [13].
- We analysed the trajectories that reached the furthest south and did not find obvious common features. However, it is not a key-part of this study to analyse extreme cases. This study focusses on the average states (e.g. average difference that is induced by the forcing). The statistics of iceberg forcing, drift, melt and occurrence are created by seeding icebergs at random dates and (to an extend) positions, introducing a random

selection of environmental conditions. Those statistics likely include extreme forcing conditions and extreme iceberg extend, but aim at representing the mean state.

- Iceberg 2013-788 is a case of exceptional southward drift, but not due to extreme forcing. It is discussed in the manuscript, why the trajectory is shown, despite its incapability of representing an average state.
- No additional deterioration processes are added to the iceberg model, as this study restrains from analysing the effect of the models settings in the simulation results (see answer [4]).
- We suggest to review the discussion (of the example trajectory) to make it more clear to the reader, that we do not draw conclusions from the example trajectory, but that it only illustrates found conclusions. we also suggest once again to highlight the scope and limitations of the study more clearly.

*8. Mechnism analysis: I would also love to see more mechanism anslysis about wave erosion, basal melt, and buoyant convection since it exactly shows how iceberg react to different forcings. This would clarify how icebergs respond to different forcing conditions. Additionally, could you explore ways to incorporate the uncertainties related to solar radiation, calving, and wave interactions, which are currently simplified in the model?*
[16]

- We initially studied the reaction of the icebergs to different forcing concerning melt processes in more detail. However, we decided to not present it in the manuscript, as the results were not very clear, did not contribute to the conclusions and the page limit was reached. A shortened version of the findings is still presented in Section 3.2.
- Wave erosion and calving are complex mechanisms that deserve dedicated studies. The interaction of iceberg and waves is under current study by other researchers, and is outside the scope of this manuscript. Other iceberg models assimilate wave data and simulate wave erosion in a more complex manner, making those models more suitable for the task.  For more, this study does not analyse the iceberg model settings (answer [4]).
- Solar radiation is neglected in this case, due to the high latitudes. I suggest to make following changes:
  - Line 588: "In this study, the effect of solar radiation and forced convection by wind is neglected, due to their comparably small contribution (Savage, 2001), especially for small icebergs and high latitudes."
- A uncertainty analysis would be a very meaningful contribution. However, a uncertainty analysis concerning the melt terms of the iceberg model (and thus the model settings) is not goal of this study (see answer [4]). An analysis of the uncertainty induced by the uncertain input data is hopefully subject of future studies. This study is about the impact of forcing differences (by the model setup, resolution and also uncertainties), thus, may lead the way to this natural next step. However, I want to remind about the scope and page-limit of this study.
- We hope this comment can be resolved by outlining the scope of the study and inform about the reasoning for the model settings and analysis choices better in the reviewed manuscript.

References:

- Eik, K.: Iceberg deterioration in the Barents sea, Proceedings of the International Conference on Port and Ocean Engineering under Arctic Conditions, POAC, 2, 913–927, 2009a.
- Eik, K.: Iceberg drift modelling and validation of applied metocean hindcast data, Cold Regions Science and Technology, 57, 67–90, https://doi.org/10.1016/j.coldregions.2009.02.009, 2009b.
- Keghouche, I., Bertino, L., and Lisæter, K. A.: Parameterization of an Iceberg Drift Model in the Barents Sea, Journal of Atmospheric and Oceanic Technology, 26, 2216 – 2227, https://doi.org/10.1175/2009JTECHO678.1, 2009.
- Keghouche, I., Counillon, F., and Bertino, L.: Modeling dynamics and thermodynamics of icebergs in the Barents Sea from 1987 to 2005, Journal of Geophysical Research: Oceans, 115, https://doi.org/10.1029/2010JC006165, 2010.
- Kubat, I., Sayed, M., t, S. B., and Carrieres, T.: An Operational Model of Iceberg Drift, International Journal of Offshore and Polar Engineering, 15, 2005.
- Kubat, I., Savage, S., Carrieres, 650 T., and Crocker, G.: An Operational Iceberg Deterioration Model, Proceedings of the International Offshore and Polar Engineering Conference, 2007.
- Monteban, D., Lubbad, R., Samardzija, I., and Løset, S.: Enhanced iceberg drift modelling in the Barents Sea with estimates of the release rates and size characteristics at the major glacial sources using Sentinel-1 and Sentinel-2, Cold Regions Science and Technology, 175, 103 084, https://doi.org/10.1016/j.coldregions.2020.103084, 2020. Savage, S.: Aspects of Iceberg Deterioration and Drift, pp. 279–318, Springer Berlin Heidelberg, Berlin, Heidelberg, ISBN 978-3-540-45670-4, https://doi.org/10.1007/3-540-45670-8_12, 2001.

---

## Author Comment (AC2)

**Response to review of "Sensitivity of iceberg drift and deterioration simulations to input data from different ocean, sea ice and atmosphere models in the Barents Sea (Part II)" by Herrmannsdörfer et al.**

In the following, see numbered Reviewers comments () and Authors response [].

**General Reviewer comments:**

This is the second of a two-part manuscript. I have reviewed Part I and recommended it to be merged into Part II. (1) The Part II paper is the main body of results from the study, presenting Lagrangian iceberg simulations. Similarly to Part I, the focus of the paper is to exhibit the differences between iceberg simulations when switching one source of ice-ocean or atmospheric input data, but without an indication of the respective accuracy of each data sources.

(2) Part II simulates synthetic icebergs from their calving location until they are mostly melted, not observed icebergs. One example of simulated trajectory is selected and discussed in more details. This is an important information that should already be given upfront in the abstract to set the scene of the study.

(3) The dependency of Part II upon Part I is less strong than I originally thought. I have only counted three facts that could be included in this manuscript in replacement of some overly long technical parts (see below and see the Part I review).

(4) The model description is given in Annexes, although it is not clear whether the model is identical to the previous Monteban et al. study or if it has been modified. Either way, the authors should reconsider if the annexes need to repeat the previous paper or simply refer to it.

(5) The results are novel and interesting, and all the more important that the related literature is now quite outdated. However the Part II paper often lack a reflection on previously published results. This aspect should be strengthened.

(6) The discussion of the results could be reorganised to set the higher priorities first: if the location of the sea ice edge is the largest cause of divergence between the iceberg trajectories, then the authors could argue that the efficient assimilation of satellite sea ice concentrations is the priority issue when simulating icebergs and down-prioritise the technical issues on the search radius.

(7) The conclusion lack practical recommendations for other practitioners of iceberg modeling. Seeing that the statistics are "surprisingly similar" for the Topaz and the Barents-2.5 models, but knowing that fewer years of reanalysis are available from the Barents-2.5 model, I would be tempted to use the Topaz reanalysis all along and use the 30 years of data for the sake of statistical significance. Further, I would expect the conclusion to bring up the following topics: how important is the duration of the reanalysis, the consistency of the input data (with the example of the modifications of the Barents-2.5 model), the differences from / the commonalities with previously published iceberg simulation studies.

(8) The language should be improved with a better choice of vocabulary, logical transitions, and punctuation.

(9) Part II overall represents an interesting and valuable contribution to the field, although some improvements would be necessary, and I can recommend it publications after major revisions.

**Author's response to general comments:**

The authors thank the reviewer so much for the thorough and constructive feedback and for valuing our results. We think that we can improve the quality of the final product significantly with the help of the provided comments, hopefully leading to a successful publication. Based on the comments, we plan following major revisions to the manuscript:

- Although we put a lot of effort into part 1, we agree (after a lot of thinking) that merging part 1 and part2 into one paper will lead to a better overall product. (See reviewer's comments 1,3)
- We will re-structure the manuscript attempting to improve conciseness and coherence. Certain parts will be shortened (e.g. Section 2.5). (4,6) , and we will consider to arrange the sections of the manuscript by priority, especially the discussion (e.g. Section 4.1). (6)
- We will set the scene more clearly about what we analyse and what the limitations are. For example, we will mention that we do not include iceberg observations and that we do not analyse the impact of the iceberg model settings. (2,4)
- We agree to add a reflection on previously published results (5).
- We will add a discussion on the advantages and disadvantages of the different input datasets (e.g. temporal availability, consistency and resolution of environmental data) and the potential use for different applications (e.g. long term statistics or short-time forecast of individual trajectories). Therein, we should highlight how the choice of forcing input depends on the simulation goal. We will also investigate the projectability of our results on other studies. This shall serve as much-needed input to make educated decisions on forcing input for iceberg simulations. However, we cannot provide a generalised practical recommendation on which environmental data works best as forcing to iceberg simulations, as the suitability is highly sensitive to the application, region and time. (7)
- We will carefully consider whether or not to remove the annex (4)
- We will pay more attention to the  language and we will certainly improve it. (8)

Furthermore, we plan to change following:

- We will improve the description of similarities and differences to Monteban (2020)'s iceberg model setup. (4)
- We will discuss the potential for improvement by assimilating satellite-based sea ice concentration products. (6)

**Authors response to detailed reviewer's comments:**
Detailed reviewer's comments that we comply to with response:

- l6: "are sensitive". I would expect the abstract to indicate a degree of sensitivity, it is very or moderately sensitive? [10] Although challenging to quantitatively measure the sensitivity, we will try to indicate at least subjectively the "degree of sensitivity" as suggested.

- l12: "surprising similarities". A word of explanation is necessary in an abstract: why are they so similar? Otherwise the reader is left on an apparent contradiction with the sentence on the sensitivity of the model. [11] We will rephrase this considering your comment.

- l41: the multiplication 4.2603.7 is mysterious, but the exact number is not necessary in an introduction.
- l62: Explain why 2603*7 or remove if the exact number is not important yet.
[12] The exact number is not important. We mentioned the numbers to express that we did a statistical analysis with a reasonably large number of samples, that is typical for the region We will change this in the manuscript (in e.g L41, L62 and L537)

- l71: Why do you need geostrophic currents when the models provide surface currents? And do Slagstad et al 1990 provide data relevant for your period or only the general formula for geostrophy?
- l92: How the pressure gradient force relates to geostrophic currents is not clear. Do the authors mean to account for the sea surface slope, with a force downslope rather than turning around the positive/negative anomalies of sea level? In this case, which data is used for the sea surface heights? The reader should not have to read Slagstad et al. 1990 to understand this sentence.
- l570: I was hoping to understand what "pressure gradient" is meant in the Annex, but to no avail. The confusion between ocean currents and geostrophic currents is still puzzling and the notation Fc,p rather indicates that the pressure gradient is the same as the Coriolis force.
[14] We inherited the model setup including drift equations from Monteban (2020). We found a typo in Equation A10. We correct the equation to: $F_{c,p} = m \cdot f \cdot [v - v_{geo}; -u - u_{geo}]$. It is expressing Coriolis and pressure gradient force in iceberg drift simulations in one term. The idea is to subtract the geostrophic velocity of the ocean for geostrophic balance to receive the "Coriolis-related term" (Gladstone 2001 and used by Stern 2016, Monteban 2020). Slagstad (1990) only provides a general estimate. Nowadays data availability allows for including the sea surface height and slope more accurately. However, we do not intend to study the impact of the model setup

- l79: "Gaps of [...] few days", how many days? Winds are unlikely to persist for more than two days in the Barents Sea. [15] *"gaps on the scale of hours to 7 days"*.

- Section 2.5: Distinguish assimilation from forcing.
- l44: Forcing a model is not data assimilation (systematic confusion, also commented in Part I).
[16] We are aware of the difference between assimilation and forcing. In the context of this work, we use the terms:
  - Assimilation/to assimilate: The iceberg model takes in input data (with a certain method).
  - Forcing- variables, data, data set (sometimes just "forcing"): Input data
  - Forcing fields: 2d/3d/4d input data
  - (nearest) forcing (grid) cell: Grid cell of the input data, that is closest to the current iceberg position. It is used as part of the assimilation method.
We did not find any other terms containing "assimilation" or "forcing" in Section 2.5 and did not find mix-ups between those terms. Nevertheless, we will pay extra attention to these definitions and try to avoid any confusion in the revised manuscript.

- l107: "for one time step": Time should be irrelevant for spatial interpolation. [17] "Timestep" refers to iceberg simulation time steps and its position. We will change to: *"Note that the environmental data sets have different grids, so that the forcing data for the same iceberg position is not raised from the same area."*

- Section 2.5 has too much details, and Figure 1 is not necessary since the ocean currents are notably inaccurate near the coast, as noted by the authors much later. The takeaway from that

section is that inputs near the coast are fetched from a nearest neighbour, which can be several grid cells aways. As a reader I am ready to accept that there are too many uncertainties near the coast and that as long as icebergs are seeded to the ocean, the following simulation is valid.
- Section 3.1 is similarly too detailed and Figure 2 can be removed without affecting the following results.
[19] We plan to shorten Section 3.1, 4.1 and 2.5 to the key-message to enhance the readability of the paper.

- Table 2: Why not indicate total mass loss? The relative contributions can be misleading when one model melts icebergs much faster than the other. [18] We can include total mass loss mentioned in line 189 into Table 2. The relative contributions can certainly be confusing, but they also help to explain why and how the total mass loss varies. Thus, we will keep the relative contributions, unless strongly opposed.

- l179: Is M_fb the same as M_fw? [20] Yes, we will correct the typo.

- l201: What are "seeding characteristics"? The number or the size of icebergs seeded? An explanation why icebergs from some glaciers last longer than others would be interesting. [21] We add "seeding characteristics (number and size)..." in L201. We also add to the discussion 4.2 that icebergs from Franz-Josef-land last longer due to their larger seeding size and frequent presence of sea ice around the archipelago, while icebergs from Novaja Zemlja and Edgeøya are initialised with smaller size and in warmer, mostly sea ice-free waters.

- l210: that the drift distance is longer with Topaz is surprising, I would have expected that the tidal loops would have made them longer with Barents2.5. [29] We also found it surprising and could explain the behaviour for those sources.

- l223: In which projection is defined the output grid? [22] The output grid for the iceberg density resembles a curvi-linear Topaz grid at reduced resolution. This will be added in L223.

- l230: Are there other results in the literature to support these results? [23] We plan to relate the results to previous findings. However, we should not set focus on it, as we do not study absolute iceberg density, but the differences in density due to varied forcing.

- Figure 4 is too small for the paper version, but by blowing up on screen, the Barents2.5 shows more fine features than Topaz, which may be related to topographically steered currents. A few isolines of the topography (50 m or 200 m) may help reading this map. If Figure 4 is made bigger and more readable, Figure 5 can be removed as it seems redundant. [24] We agree that Figure 4 needs to be enlarged and that bathymetry isolines would help in the interpretation. However, we strongly advocate to keep Figure 5, as it i) contains some of the key-findings of this study and ii) makes the comparison of densities significantly easier than comparing the maps in Figure 4.

- Section 3.5: The notion of iceberg extent was introduced before (Keghouche et al. 2010, possibly earlier), is the same definition applied here?
- l448: About iceberg extension and environment forcing, Keghouche et al. 2010 did show the relationship between iceberg extent and the wind patterns. Does that relationship still hold in your study?
[25] Keghouche (2010) was certainly an inspiration for this study. We should mention Keghouche (2010) and relate our findings with the described dynamics and thermodynamics of interannual variability in iceberg extension. However, the studies are only comparable to a limited extend due to different focus of the analyses.

- Table 4: Sea water surface velocity is the norm or the v-component of the velocity?
- Table 4, the column named v_ai does not correspond to the caption. It seems the water and sea ice velocity columns have the wrong name.
 [26] Table 4 provides the norm (the mean total sea surface velocity, that the iceberg experienced along its trajectory). The same is true for wind and sea ice drift. The variable notation is introduced in Section 2.2. We add "total [...] speed" in the caption to Table 4. The typo "v_ai" will be corrected to "v_si".

- Table 5: Why are there two numbers in the last three columns? [27] The first number of each column shows the values along the entire trajectory, while the second number shows the values for all times with relevant sea ice. As the second numbers are not used in this version of the manuscript, we remove them.

- Figure 9 panel a) add the horizontal line at 0 degrees to indicate the melting temperature of glacial ice. Panels d-e-f are not very informative as they show the velocity modulus but not the direction. Feather plots could be more intuitive. [28] We agree to add the 0degree line to panel a). We think that panels d-f are valuable, as iceberg drift and deterioration largely depend on those speeds. We wanted to show the different temporal variability in the different forcing data, due to their temporal resolution and e.g. tides. We did not add feathers, as the temporal variability is too high. We did not plot the direction as line plot or plot u and v component separately, as we did not use any directional information in this version of the manuscript.

- Same figure: the "delta" and "abs" on the vertical axes are mysterious, I think they can be removed. [30] The naming conventions relate to the ones used in the Table. We exchange a "delta" with "δ" (as used in Table) and exchange "abs" (==absolute) with the unit (10^5 kg2h−1). This should make the reader understand, that we no longer describe relative contributions, but absolute mass loss.

- l330 sounds rather dramatic about the data availability near the coast, but pragmatically the detailed conditions near the calving front are very uncertain anyway. Iceberg simulations may be stuck near the calving front (and need an initial push as you do with data interpolation) but as soon as they have escaped, the iceberg trajectory is valid.
- l333-340: Similarly, these considerations seem out of place, unless there is an implicit aspect that I am missing. From the results presented above, the resolution is of lesser concern than the sea ice edge disagreements between the models. The discussion could thus be shortened if it was restructured to highlight the mot important topics.
[31] The icebergs are seeded with a distance to the glacier front in which forcing data is available. However, icebergs may drift close to coastline and loose forcing information in the course of their lifetime. We found, that this happens more often for Topaz forcing due to its larger grid cell size. Due to the larger grid cell size, Topaz also provides uncertain information further into the sea, than Barents-2.5. The coarser land-sea-mask in Topaz further increases this effect. This considerations are only important along the coastlines and regions with complex bathymetry/topography. We agree that the described effects are less important than the sea ice representation and agree to shorten the section and shift it to a later point of the discussion. We also express more clearly where this effect is relevant while also stressing that the forcing is highly uncertain along the coastline anyways.

- l365: Although I expect this sentence to be correct, I fail to see this increased deterioration in May-June in Figure 9h. [32] We will double-check if Figure 9h is plotted correctly and change the

text if necessary. We should also mention that the deterioration rate (in kg) decreases with decreasing iceberg size.

- l372: The difference of temperature between Barents-2.5 and Topaz only comes here. The differences in sea ice variables come further on. This means that the relevant elements of the Part I paper can be inserted before section 4.2. [33] While it is true that the comparison of Topaz and Barents2.5 is mentioned first in those lines, we think that the comparison would be better integrated in another place for the sake of the reading flow (See reponse [1]).

- l375: The Barents-2.5 may well have too cold SST, but a comparison to satellite data would hammer the facts. [34] Including a comparison to satellite data is entirely out of the scope of the "merged" manuscript. Therefore, we use known error from e.g. Röhrs (2023) (see response [1c]).

- l381: The tidal loops should be mentioned earlier when the datasets are introduced. I cannot see tidal loops in Figure 7, are they too small for the figure or are the tides small in that area? [35] We will mention the representation of tides in the comparison of Topaz and Barents2.5 that will be adapted from part1. Iceberg 2013-788 is influenced by tides, as shown in the oscillation of the Barents2.5 sea water speed along the trajectory in Figure 9d. However, the tidal forcing along the trajectory forced by Barents2.5 seems to be too small to be visible on the scale of Figure 7. Other iceberg trajectories showed obvious tidal loops, but we decided to not include more examples.

- l385: The tidal component is essential for iceberg density and extent. This statement is not obvious from the figures, nor followed up in the rest of the paper. Could it be elaborated? [36] We should add that the tidal looping causes the iceberg to spend more time in the region and ii) thereby keeps the iceberg from drifting into warmer waters, which increases the drift duration and ii) increasing the local iceberg density (more timesteps with icebergs in density-grid-cell). We should elaborate in l385: *"One might conclude that the tidal component is not essential to where icebergs mainly drift in the Barents Sea (main pathways), however we found that it is essential to simulating individual iceberg trajectories, how many icebergs drift in different regions of the domain (iceberg density)."*

- l391-394: I miss the whole idea of this paragraph. Please rephrase to clarify. [37] The idea of this paragraph is to explain that regional differences (e.g. between Franz-Josef-Land and Hopen Trench) dominate ocean mode differences (e.g. SST difference in Barents2.5 and Topaz). We conclude that the ocean models capture the regional differences despite their differences, which is an important message for the choice of forcing input for iceberg simulations. We will rephrase the paragraph.

- l408: This sentence is inconclusive about the mobility of icebergs near the calving glaciers, their grounding and data interpolation near the coast. Keghouche et al. 2010 computed the incidences of grounding, could the authors compute a similar map or a blowout of the averaged current vectors near a calving front to discriminate which effect is most important? [38] See answer [31], where we describe that icebergs did not ground close to their seeding location, but in their later drift. We, in general, observed little grounding events in comparison to timesteps in which forcing was not available (in the nearest forcing cell). Grounding events may have been more important in Keghouche (2010),  due to different iceberg seeding sizes and potentially smaller melt rates due to different sea ice, ocean and atmospheric properties during the years 1985 to 2005.

- l425: "might increase": this sentence is blue sky to me. More sea ice means less melting or wave erosion, and additional sea ice stress, so the authors could be more assertive about the difference between the models. Also mention which model is more realistic. [39] We rephrase: "The more extensive sea ice cover over the Spitsbergen Bank in Barents-2.5, and thereby reduced melt rate by wave erosion and buoyant convection, increases the number of icebergs drifting as far south as Hopen Trench." As we did not find any previous studies comparing Topaz and Barents2.5 (to the same set of observations), we cannot say which model is more realistic with confidence. However, we try to mention weaknesses and strengths of both ocean models that may help in the judgement of which forcing works better in specific applications of iceberg simulations.

- l430-431: This sentence is too complex. Do you mean that higher resolution winds follow better the orography? [40] We wanted to say that higher resolution wind is more accurate over complex orography. We found that the addressed sentence is a repetition and therefore remove it.

- l432: The coarse resolution currents are extrapolated near the coast, irrespective if the currents are on-shore or off-shore, so I don't understand why the icebergs cannot drift close to the coast in Topaz. [41] We think that the addressed sentence creates more confusion than it adds information and remove it.

- l445: "iceberg drifted": it is singular because there was only one iceberg? [41] It should be plural ("icebergs").

- l451: I don't understand what is meant by varied forcing reproducing the variability. Rephrase. [43] "A similar multi-year variability of iceberg extension is simulated independent of the forcing input."

- l466: This idea is repeated. [44] We remove addressed repetition.

l470: Has the influence of the seeding mechanism been accounted for in the presented results or only mentioned as a warning to the readers? [45] The influence of the seeding mechanism has been accounted for in the discussion, but is still present in Figure 6. We remove the sentence to avoid confusion.

- l472-477: It is well known that Lagrangian trajectories diverge over time (there is a vast body of literature about dispersion in the ocean surface since Okubo 1971, see also Koszalka et al. 2009 for a geographically closer example), the first part of the paragraph is correct, but the end is unrelated: one single example of iceberg trajectory does not say much over averaged statistics (See Figure 4 for example).
- l475: "Similar initial conditions". Are they similar or strictly identical?
- l479: Use a more precise vocabulary: The iceberg trajectories diverge.
[46] We will highlight that the example trajectory is only presented for illustrative and explanatory purposes and that it cannot be used to make broad conclusions. We should also make sure that the reader is not confused into thinking we (wrongly) based our conclusions on known divergence of Lagrangian trajectories. We will change to" *The sensitivity of iceberg simulations to their environmental forcing is illustrated by the example of iceberg 2013-788. The example demonstrates how identical initial conditions and small deviations in the environmental forcing can still diverge (causing further deviation in forcing). Different drift trajectories ultimately lead to different potential exposure of structures and ships to icebergs.*"

- l483: Here again, the end of the paragraph seems unconnected from the preceding argument. [47] *".. This also highlights the importance of temporal and horizontal resolution of the forcing data. Due to its high horizontal and temporal resolution, the use of Barents-2.5 may be beneficial in iceberg simulations, compared to the lower resolution Topaz data."*

- l496: "can be seen", are you referring to Figure 9? [48] Yes, we add "in Figure 9" in l 486.

- l489: Sentence unclear. It seems to be rephrasing the same idea, but with unclear words ("derive"). [49] We change to: "This illustrates how the sensitivity of iceberg simulations to sea ice forcing is given by the large impact on drift and deterioration and the large occurrence of sea ice in the iceberg pathways.

- l499: You found that the results were not so sensitive to the change of atmosphere reanalysis, do not state the opposite in the conclusions.
- l500: Surprising similarities: Please elaborate on how the differences in the forcings cancel out in the final statistics.
[50] We want to communicate the small difference due to the atmospheric forcing that we found (e.g. in iceberg densities). We don't want to go into detail in the introductory paragraph of the conclusion. The similarities are discussed in l521-526. We change to: *"We found that the results of such simulations are sensitive to the input from ocean, sea ice and atmosphere reanalyses or forecasts. However, the extend of the impact varied with the environmental parameter and the iceberg characteristics.*

- l502: This statement is contrary to previous statements that average sea ice thickness and sea surface temperatures cause most of the differences, not so much the resolution of the data. The resolution only seems to be an issue near the coast.
- l503: the enumeration lacks logic. This paragraph is restating previous results instead of concluding on the take-home message.
[51] We agree that we should be more precise but also conclusive in this paragraph. We did not intend to say that the resolution has larger impact than e.g. the sea ice concentration. We change the paragraph to the below and shift it further back (e.g. in front of the similarities paragraph): *"Spatial and temporal resolution of the forcing data cause large impact in simulated individual trajectories (e.g. the location and timing of when an iceberg drifts out of the sea ice). The horizontal resolution has a small impact on the iceberg statistics (e.g. the iceberg density in the domain) by influencing the availability of forcing information and its representability for the iceberg position, along the coastlines and regions with complex bathymetry. We highlight the importance of this effect in coastal regions, despite its unreliable forcing information, due to the lack of other (environmental and iceberg) information."*

- l515: Did we see that the iceberg looping increases their density? [52] We will add it to the discussion as described in answer [36]. We think that is appropriate to be mentioned in the conclusion as well.

- l572: Drag coefficients are different from those in Table 2. [53] The coefficients in Table 2 are correct. "and water and air drag coefficient $C_{w,a} \approx$ ."

We will change following as suggested and no further comments are needed. We also adapt the manuscript for the comments about typos and language as suggested.

- l19: "ice features" -> "icebergs"
- l20: Grounding is missing here, and only mentioned far down in the manuscript.
- Grounding is missing from Section 2.4 and first mentioned on l.412.
- Table 1 could contain more information in additional columns and remove Column 1 (objective) which is not followed up in the text: for example, the duration, horizontal resolution, frequency and presence of tides.
- l66: the acronyms ERA5 and CARRA have already been introduced.
- l72: Indicate the version of the IBCAO bathymetry used.
- l82: Many readers may be unfamiliar with the area, move the general map A1 from the annex to this location. Hopen, Bear Island and Storfjorden should be indicated as well and a few isolines of the ocean bathymetry.
- l85: "Empirical relations". Indicate the sources of these relations
- l94: "Coefficient set to zero". Indicate "no added mass" because the coefficient has not been defined here.
- Figure 9 g-j continues on a different page, so the related part of the caption should follow as well.
- Figure 6 is not colourblind-friendly. I cannot tell the two CARRA simulations apart. Perhaps make them dashed lines.
- l251: Contradiction between the two sentences. Replace "all directions" by "most directions"?
- Section 3.6: add "simulated" in the section title for clarity.
- l397: "may indicate an impact". Please introduce the next discussion in a more direct way.
- l491-495: This paragraph sets the context of the results and should be moved to the beginning of Subsection 4.4
- Eq. A15. Indicate the reference for this equation.
- Section A5. It is not necessary to have a section header and one line there since the Tables are referenced in the text.

References:
Stern, A. A., A. Adcroft, and O. Sergienko (2016), The effects of Antarctic iceberg calving-size distribution in a global climate model, J. Geophys. Res. Oceans, 121, 5773–5788, doi:10.1002/2016JC011835.

---

## Author Response (AR1)

**Author's response to the reviews of "Sensitivity of iceberg drift and deterioration simulations to input data from different ocean, sea ice and atmosphere models in the Barents Sea (Part II)"**

We again thank the reviewers and the editor for the valuable input. We think that, after the applied major revisions, the manuscript has increased in quality significantly and think that it's publication will be a valuable contribution to the field.

The authors already provided a detailed response on the reviewer's and editor's comments in the interactive discussion. All comments were considered, and the suggested changes were applied to the manuscript. The highlighted changes can be found in the Author's track-changes file.

The main changes in the manuscript are:

- Laying out the "story-line" of the study more clearly by restructuring, deleting and adding sections. We revised the manuscript (especially the discussion) on cohesiveness.
- Explaining the goals, scope and limitations of the study more clearly from the beginning and managing the readers expectations. We also highlight the novelty and importance of the study.
- Including a short comparison of the environmental variables from Part1 and using the knowledge in the discussion. Corresponding to the review of Part1, we explained more carefully how we carried out the comparison, its limitations and how it connects to earlier findings.
- Providing clear results and take-away-messages.
- We also added a discussion of the suitability of the environmental datasets as input to iceberg simulations and recommendations of their use in different applications.
- We also improved the language and revised all figures and tables.

If the reviewers or the editor require a detailed list of changes, the authors can certainly provide one. We are also happy to receive a second round of reviews.

---

## Author Response (AR2)

Authors response to second review of "Sensitivity of iceberg drift and deterioration simulations to input data from different oceans sea ice and atmosphere models in the Barents Sea" by Herrmannsdörfer et al.

(Review in black text colour and authors response in blue text colour)

The reviewers have complied to my main objections on the previous iteration (merging the two parts of the paper), provided new figures and improved the English grammar. Some other suggestions or criticism have not been taken up and - since no point-by-point rebuttal has been provided - I ignore why the authors disagreed with me. Some of my comments below will therefore be repeated from the first review, but I do not see any other solution.

I am however positive that the article is almost ready for publication in The Cryosphere, with the exception of minor requests listed below: There remains some repetitions in the text that the authors should chase for the sake of conciseness and the use of "geostrophic currents from Slagstad et al. (1990)" seems to be a misunderstanding. Also parts of the Appendices may be superfluous if the iceberg model is mostly unchanged from the literature.

The authors appreciate that the applied changes are positively recognised and that the manuscript is considered almost ready for publication. We apologise that the uploaded response document (in the interactive discussion section "AC2: Reply on RC2") did not explain and describe the changes detailed enough. We accept the suggestions of the reviewer listed below and make sure that all changes (minor comments etc.) are documented thoroughly in this review round.

**Comments on the appendix:**

- I am not convinced that the Appendices A2 to A4 are necessary if the model is identical to that of Monteban et al. 2020. If there are only a few modifications in the model, it would be sufficient to indicate these modifications in the main text. Appendix A1 could be merged in the main text where the sentences are already repeated.

We follow your suggestion to remove the appendix. For a description of the iceberg model, we refer to Monteban (2020). Only little changes in the methods section are necessary:

- L104 is changed to "The drift and deterioration equations and model parameters can be found in \citep{Monteban2020}."
- Delete L90 "More details on the seeding approach are given in the Appendix (Sect. \ref{sec:appendix})."
- Change in L81 "Following the seeding procedure in \citep{Monteban2020}, icebergs are seeded near the tidewater glaciers of Franz-Josef-Land,.."
- L68 is removed "A detailed description of the iceberg seeding, model setup, drift and deterioration equations, parameters, and computational routines is provided in the Appendix (Sect. \ref{sec:appendix})"

Aggregated comments about the usage of geostrophic currents and Slagstad data:

- l.94: I believe this sentence is incorrect: The geostrophic currents result from the equilibrium between the Coriolis force and the pressure force (down the slope of the surface heights, called pressure term Fp in Appendix A3). The geostrophic currents are therefore perpendicular to the slope of the sea surface heights.

- l.117: The currents in TOPAZ or Barents-2.5 already include geostrophic currents. By elimination I believe that Slagstad et al. (1990) provide the expression of the force exerted by the sea surface slope (also called pressure force). See comment on appendix A3 futher below.
- Formula A10 is nowhere to be found in Slagstad et al. (1990) and geostrophic currents are not mentioned in there. The authors should indicate the actual source of geostrophic currents, if any. I do not even understand why an extra source of geostrophic currents is needed since the ROMS and HYCOM models do include these currents by construction.
- l.556: As repeated from my previous review, Slagstad et al. (1990) simulated a period in 1988, but not the period of this study (2010-2021). If these model results are used in this study, they should be compared with the TOPAZ and Barents2.5 results.

Thank you for the comment. Precise notation is very much appreciated here.

You are right about Topaz and Barents-2.5 already including the geostrophic currents. The geostropic current data is from Slagstad. The data is a time averaged. Changing the model setup regarding this term is however not the interest of this study, as it is considered part of the iceberg model. This also applies to comparing the geostrophic current data to the current data from Topaz and Barents-2.5. We do not face this kind of problems in more sophisticated iceberg models and therefore a detailed analysis is not of value to the research community.

The equation is taken from Stern (2016). Following, L596 could be changed to "The pressure gradient force is approximated with the "Coriolis-related term" \citep{Stern2016} assuming geostrophic balance and using average geostrophic water velocity from \cite{Slagstad1990}. "

As the detailed description of the iceberg model is deleted with the appendix, the term and the slagstad data is explained in detail and the reader is forwarded to Monteban (2020). We try to give an overview over the term in L94. L.94 is changed to "The pressure gradient and Coriolis forces are included under the assumption of geostrophic balance." We delete L117 as it is described in Monteban (2020).

**Detailed comments:**

- Abstract l.14: add "however" or another logical conjunction to indicate the contrast with the previous sentences.

"However" was added to highlight the contrast in the sentences.

- l.42-44: the ideas of this sentence come in disorder.

We change the order and split the sentence for improved readability:

Previous version: "These environmental models describe the highly complex interaction of ocean and atmosphere of the Barents Sea due to its complex bathymetry and position between warm Atlantic waters and the cold Arctic Ocean with different resolution, model physics and representativity of the domain."

New version:" These environmental models describe the highly complex interaction of ocean and atmosphere of the Barents Sea with different resolution, model physics and representativity of the domain. The complexity of the Barents Sea arises, among other factors, from its varied bathymetry and position between warm Atlantic waters and the cold Arctic Ocean."

- l.71: The selection of data sources is very much focused on Norway / Europe. It would be fair to acknowledge that there are many other valid reanalyses that could have been used instead but

**have not.**

This is a very good point that should be mentioned.

We added following sentence in l.42: "More suitable models exist, that are not considered in this study, but could be examined in future research."

We also added following to l.553 (Conclusions): "We note that the choice in ocean, sea ice and atmosphere models exhibits a focus on European and Norwegian models, which is motivated by the location of the study area in the European Arctic and the accessibility to research purposes. We highlight the opportunity to extend this study using a larger number of suitable ocean, sea ice and atmosphere models that are disregarded in this study."

- l.91-92: This sentence is repeated from the introduction.

The sentence is varied to "The numerical model for the simulation of iceberg drift and deterioration and its settings is adopted from Monteban (2020)."

- l.97: h\_min is coming from the sea ice strength in the EVP rheology, this should be explained in the text.

We add following explanatory sentence in l.98: "Note that heavy sea ice is assumed under the simultaneous occurrence of high CI and sufficient sea ice strength. Sea ice strength is described by the thickness threshold h\_min, which is derived from an empirical relation of CI (Lichey and Hellmer, 2001)."

Also we moved the sentences about the sea ice categories in front of the explanation of the melt processes.

- l.98: "as" is falsely implying that there is no swell in the area. Rather make the assumption that the swell has little effect on the erosion of small icebergs.

We clarified the statement as follows: "The wave erosion term disregards swell waves due to simulated small iceberg sizes but includes local wind waves as function of wind and water velocity."

- l.106: "Assimilates" should rather be replaced by "is driven by" as no data assimilation is performed in this study. Similarly in l.109 and 110.
- "Assimilate" is replaced by "driven by" and "taken from".
- l.137: The sea ice edge is not identical in Barents-2.5 and Topaz. Which of the two is used here?

The average sea ice edge of both models are depicted in the Figure. The CI difference is calculated where one or both models show CI>15% at any point of the study time. We changed l. 137 to following: "..within the maximum modelled sea ice extent, including areas where sea ice is present only in one of the models and during limited time periods."

- Table 2: The 10m winds appear twice in the table, both in the ocean and in the atmospheric models, with different values and with the opposite difference between ERA5 and CARRA. Why? Are they computed on different domains?
- Figure 2: Specify whether the differences are TOPAZ-Barents and ERA5-CARRA, consistently with Table 2, or the opposite.

Wind speed differences are computed on different subsets of the domain and time period. Wind speed differences are calculated along all trajectories using ERA5 (Combinations Topaz-ERA5, Barents-ERA5) and CARRA (Combinations Topaz-CARRA, Barents-CARRA) (Table2, atmospheric models). This shows the difference due to the wind input. The wind speed difference is also

calculated along all Topaz trajectories (Combinations: Topaz-ERA5, Topaz-CARRA) and Barents-trajectories (Combinations: Barents-ERA5, Barents-CARRA) (Table 2, ocean models). This shows how the wind varies, not due to the wind input, but due to trajectory difference due to ocean and sea ice input.

To explain the difference more thoroughly we add and change following:

- Add in Fig2 caption: "The values for v\_a are given for the iceberg pathways as simulated by different ocean and sea ice (a) or atmospheric (b) input." And added a,b in the Table.
  -change l.171 to "..and varies just as much between the two atmospheric models as between the pathways caused by varied ocean and sea ice input."
- l.157 throws many issues that may cause low predictive skills without explanations. The authors should either explain or remove this sentence altogether.

  We agree to remove any explanation for low predictive skills, as it is outside of the scope of the study and the authors' experience. We change the sentence to "Model skill varies over time and spatial scales and the predictive skill for surface water speed and direction in Barents-2.5 is low. Some skill can be.."
- l.180. Should we expect that other models exhibit larger differences?

  Yes, we may expect larger differences for models that are in no way connected. However, that does not exclude the possibility that some "un-connected" models show high similarity.

  We add the sentence "Unrelated atmospheric models may exhibit larger differences."
- l.200: Is 62 tons the mass loss for one iceberg? From which initial mass?
  62 tons is the average massloss for a iceberg from simulation start to end. The initial mass varies strongly with random iceberg size seeding. Iceberg size seeding is reproduced exactly for simulations with different environmental input (e.g. Barents), meaning that the deterioration difference is caused solely due to the environmental input. We change l.200 to "The average iceberg mass loss (from simulation start to end) is larger in simulations with Topaz input (+6.2 10^4 kg)."
- l.267: Simulated icebergs are seeded from July to December. How realistic is that model setting? A sentence or a reference would be welcome.

This choice is based on the assumption that the termini of the glaciers are surrounded by sea ice (and icebergs are locked in the sea ice) for the rest of the year. This might be realistic for the icebergs originating from Franz-Josef-Land, but less realistic for Svalbard and Novaja Zemlja. We inherited this assumption as part of the seeding settings from Monteban (2020) who based the seeding settings on a large number of satellite observations he made.

- l.310-311: I don't understand this sentence, nor what it implies for the results. Are there more differences of winds because the changes of currents steer the icebergs onto different trajectories than the change of wind forcings fields themselves?

Precisely! This is the same as in Table 2, but this time for the example of one iceberg. This implies for the results that varied atmospheric input has low impact on the iceberg simulation results (compared to ocean/sea ice input).

We change L310-311 for increased understandability: "Further analysis (not shown) obtained that the wind speed varies less between the atmospheric input than between the trajectories caused by varied ocean input."

- l.319: "Further analysis"... Unfinished sentence. Thank you. We delete the phrase.

- Figure 8 is missing the colour scales for temperature. It is also a busy (too busy) figure so the empty star is hard to find and I could not see the ice edge in the right column. You could use a full star on all panels (no risk of confusing the stars), remove the bathymetry, remove wind and currents on heavy sea ice (since they are not acting anyway). This should improve the clarity.

Thank you for those suggestions. Following changes are made:

- "Empty stars" are replaced by filled black stars.
- The sea ice edge in the right column was displayed in a dotted style and is changed to a line style.
- The colour scale for the temperatures is added.
- The bathymetry was added on request in the first review round, but is removed here for improved visibility.
- Wind and current arrows are removed over heavy sea ice.
- For consistency, the similar hatches for heavy sea ice are used in both columns.
- The legend and caption are changed accordingly.
- l.374: "need to be viewed in the light of low reliability of the data". Did you mean "are more dubious/uncertain"?

Yes. "Low reliability" is changes to "large uncertainty".

- l.422: Unclear sentence. Is this the same idea as in l.310-311? I believe it can be removed without loss of information or logic.

Yes, it is the same idea as in l310 and is therefore removed.

- Table A1: Indicate units for location and scale (meters?).

Thank you. I added the unit, which is in this case meters.

**References and typos**

- l.14: "inputs" plural because there are ocean, sea ice and atmospheric terms. Correct, changed in the reviewed version of the manuscript.
- l.42: Schyberg et al. Is missing a year. Added year in reference.
- l.47: "related to" could be replaced by "extracted from". Replaced by "Extracted from".
- l.47: what is a "composite of knowledge"? "Composite of knowledge" replaced by "Combinations of knowledge". This refers to the combination of knowledge from previous quality reports and analysis done is this study.
- l.57: is missing a closing parenthesis. ")" added.
- l.60: "analysing on" -> "an analysis of" Changed to "We emphasise that this study focuses on the impact of the choice of environmental input data on iceberg statistics rather than an analyses of the absolute iceberg statistics."
- l.65: "assess" -> "to assess" Changed.
- l.131: There are two papers "Röhrs et al. 2023", please number them by a/b for disambiguation. Added "b" in notation of second paper.

- Table 1: TOPAZ temporal resolution, remove "to monthly" since only daily files are used in this study. Changed.
- Table 1 could also mention that TOPAZ data are ensemble averages of 100 members. Thank you for this information. Added to table.
- l.150 use "faster" rather than "larger" for velocities. Also in l.154. Changed.
- l.152 Missing space at beginning of sentence. Space added.
- Table 2: What does the Ø symbol mean? An average? Yes. Added "Ø denotes the variable average" in Table caption.
- l.158: "the speeds" of currents or sea ice drift? Added "water and sea ice" for clarifying which speeds are ment.
- l.161: "it's" -> "its". Changed.
- l.168: "varies just as much as in the pathway differences by the the ocean and sea ice input". Unclear sentence, can you rephrase it? Was changed to "..varies just as much between the two atmospheric models as between the pathways caused by varied ocean and sea ice input."
- l.204: "daysduration shorter" -> days shorter. Changed.
- l.224: "every occurring i" -> "every iceberg i". Changed.
- l.243: "in father" remove "in". Changed.
- l.260: "adapt" -> "adopt"? Changed to "We adopt and modify the definition of the iceberg extension from Keghouche (2010) and show the relative number.."
- l.298: use "warmer" rather than "larger" for temperatures. Changed.
- Caption of Figure 8, 5th line, the closing parenthesis should be before "resolution", not after. Changed.
- l.492: Missing REF. Reference added.
- l.503: "the year of 2010 to 2014" -> "the years from 2010 to 2014". Changed.
- l.504: "constriction" -> "constrain" Changed.
- l.507: "the basis" -> "assimilated" for once that this is the correct term. Changed to "...although these are assimilated into the sea ice models already (e.g. Topaz, Barents-2.5 Forecast)."
- Equations A15, A16 and A17 should indicate units. Added "The melt terms M are given in m s-1."
- l.632: "Xie J." -> "Xie". Changed.
- l.647: Indicate co-authors of Giusti et al., consistently with the journal standards for citations. Also punctuation error "reanalysis.," Corrected punctuation. Changed author to "ECMWF" as current version of document can no longer be traced back to "person" author.
- l.663: Is there an update of "Idzanovic et al. in progress, 2024"? The authors of the report confirmed that there is no updated or published version of the report.

- l.674: Apparent mistake in the authors list with "t". "T., and Wang, Z..." is a continuation of the reference "Schyberg et al. 2023". The print-ready version of the manuscript will have different page breaks, eliminating the problem.